# BopN is a Gatekeeper of the *Bordetella* Type III Secretion System

Kevin Munoz Navarrete,[a] Ladislav Bumba,[b] Tatyana Prudnikova,[c] Ivana Malcova,[a] Tania Romero Allsop,[a] Peter Sebo,[b] Jana Kamanova[a]

[a]Laboratory of Infection Biology, Institute of Microbiology of the Czech Academy of Sciences, Prague, Czech Republic
[b]Laboratory of Molecular Biology of Bacterial Pathogens, Institute of Microbiology of the Czech Academy of Sciences, Prague, Czech Republic
[c]Faculty of Science, University of South Bohemia in Ceske Budejovice, Ceske Budejovice, Czech Republic

**ABSTRACT** The classical *Bordetella* species infect the respiratory tract of mammals. While *B. bronchiseptica* causes rather chronic respiratory infections in a variety of mammals, the human-adapted species *B. pertussis* and *B. parapertussis*$_{HU}$ cause an acute respiratory disease known as whooping cough or pertussis. The virulence factors include a type III secretion system (T3SS) that translocates effectors BteA and BopN into host cells. However, the regulatory mechanisms underlying the secretion and translocation activity of T3SS in bordetellae are largely unknown. We have solved the crystal structure of BopN of *B. pertussis* and show that it is similar to the structures of gatekeepers that control access to the T3SS channel from the bacterial cytoplasm. We further found that BopN accumulates at the cell periphery at physiological concentrations of calcium ions (2 mM) that inhibit the secretion of BteA and BopN. Deletion of the *bopN* gene in *B. bronchiseptica* increased secretion of the BteA effector into calcium-rich medium but had no effect on secretion of the T3SS translocon components BopD and BopB. Moreover, the Δ*bopN* mutant secreted approximately 10-fold higher amounts of BteA into the medium of infected cells than the wild-type bacteria, but it translocated lower amounts of BteA into the host cell cytoplasm. These data demonstrate that BopN is a *Bordetella* T3SS gatekeeper required for regulated and targeted translocation of the BteA effector through the T3SS injectisome into host cells.

**IMPORTANCE** The T3SS is utilized by many Gram-negative bacteria to deliver effector proteins from bacterial cytosol directly into infected host cell cytoplasm in a regulated and targeted manner. Pathogenic bordetellae use the T3SS to inject the BteA and BopN proteins into infected cells and upregulate the production of the anti-inflammatory cytokine interleukin-10 (IL-10) to evade host immunity. Previous studies proposed that BopN acted as an effector in host cells. In this study, we report that BopN is a T3SS gatekeeper that regulates the secretion and translocation activity of *Bordetella* T3SS.

**KEYWORDS** BopN, *Bordetella*, gatekeeper, type III secretion system

Address correspondence to Jana Kamanova, kamanova@biomed.cas.cz.

The authors declare no conflict of interest.

The closely related classical *Bordetella* species, *B. pertussis*, *B. parapertussis*, and *B. bronchiseptica*, cause respiratory infections in mammals. The strictly human-adapted pathogen *B. pertussis* is the main causative agent of whooping cough or pertussis, a highly transmissible and acute respiratory infectious illness that has recently resurged in vaccinated populations (1). *B. parapertussis*$_{HU}$ causes a milder form of whooping cough in humans, and its other lineages infect ovines. In contrast, *B. bronchiseptica* has a broad host range and infects a variety of mammalian species, eliciting pathologies ranging from typical chronic respiratory infections to acute illnesses, such as the kennel cough in dogs, atrophic rhinitis in swine, snuffles in rabbits, and bronchopneumonia in guinea pigs (2, 3). The 3 classical *Bordetella* species produce numerous virulence factors involved

in colonization of host airways, immune system evasion, and transmission to new hosts. These include adhesins and toxins, such as the highly conserved adenylate cyclase and dermonecrotic toxins (4). A type III secretion system (T3SS) was shown to specifically contribute to the persistence of *B. bronchiseptica* in the host respiratory tract (5–7), whereas the role of the T3SS in infections by *B. pertussis* remains to be clarified (8).

The T3SS of bordetellae is a multicomponent protein-export apparatus that injects effector proteins BteA and BopN directly from the *Bordetella* cytosol into host cells (9, 10). The process requires the assembly of a pore in the host plasma membrane, formed by the two *Bordetella* translocator proteins BopB and BopD (11, 12). Further, it depends on one of the most abundantly secreted T3SS substrates, the Bsp22 protein that is thought to act as a unique tip of *Bordetella* injectisome connecting the T3SS needle and the translocon pore (13). Upon injection into host cells, the 69-kDa BteA protein, also known as BopC, localizes to the lipid microdomains (rafts) at the inner face of the cell plasma membrane and triggers a rapid death of eukaryotic cells via an unknown mechanism (9, 14–17). Interestingly, the BteA protein from *B. pertussis* exhibits a much lower cytotoxic activity than BteA of *B. bronchiseptica*, due to the insertion of an additional alanine residue at position 503 (A503), which most likely represents an evolutionary adaptation of *B. pertussis* to the acute infection lifestyle in the human host (18). The other T3SS-injected protein, BopN, has been reported to interfere with *B. bronchiseptica*-induced NF-$\kappa$B signaling, provoking upregulation of the anti-inflammatory cytokine IL-10 and thereby undermining host protective immune responses (10, 19). In addition, BopN has been described to promote BteA-mediated cytotoxicity of *B. bronchiseptica* on rat L2 pulmonary epithelial cells but not on mouse DC2.4 dendritic cells (10, 19). No mechanistic details of the BopN action(s) are known. However, the amino acid sequence of the 365 residues of the BopN protein shows 19% identity with the *Yersinia* protein YopN, which regulates the biogenesis of T3SS and belongs to the family of SctW proteins known as gatekeepers (6, 20, 21).

T3SS biogenesis in Gram-negative bacteria is a tightly regulated process in which substrates are secreted in a specific order. After Sec-dependent assembly of the basal body, type III-mediated secretion begins with the release of the early substrates that form the inner rod and needle, followed by the transport of intermediate substrates that include the needle tip and translocator proteins, and finally, secretion of the late substrates, the effector proteins (22, 23). The effectors are translocated only when the needle tip senses the host cell. However, contact with the host cell can be artificially mimicked by various chemical signals, such as low concentration of calcium ions in *Yersinia* and enteropathogenic *Escherichia coli* (24–26), pH shifts in *Salmonella* pathogenicity island 2 (SPI-2) T3SS (27), Congo red in *Shigella* (28), and high potassium ion concentration in *Vibrio* (29), all triggering effector secretion. Proteins that prevent the premature release of effectors under secretion-restrictive conditions (before contact with the host cell or without an artificial trigger) serve as gatekeepers and are referred to as SctW in the unified nomenclature of Sct proteins. The members of this family include SepL in enteropathogenic *Esterichia coli* (EPEC), MxiC in *Shigella*, InvE and SsaL in *Salmonella*, CopN in *Chlamydia,* and previously mentioned *Yersinia* protein YopN (22, 23). These proteins interact with the SctV type of proteins of the T3SS export apparatus, and it has been suggested that they block secretion of effectors either by forming a physical barrier, or by having allosteric effects on SctV function, while supporting the loading of translocator-chaperone complexes (30–33). Following an activation signal (contact with the host cell and/or an artificial trigger), the gatekeeper is released from SctV and effector secretion is initiated (31, 33). Inactivation of the gatekeeper causes deregulation of effector secretion independently of activation conditions and accounts for defects in targeted effector delivery. For example, *Yersinia* mutants lacking the gatekeeper YopN secrete effector proteins even in the presence of 2 mM $Ca^{2+}$ (secretion-restrictive conditions) and display leakage of effectors into cell medium during cell infection (34–37). Interestingly, the fate of the gatekeeper after its release from SctV varies among bacterial species. It is either degraded, as in the case of the SsaL

gatekeeper of *Salmonella* SPI-2 (27), or it is translocated into host cells, like YopN of *Yersinia*, MxiC of *Shigella flexneri*, or CopN of *Chlamydia* (36, 38, 39). Gatekeepers are not thought to have effector functions when injected into the host cell, except *Chlamydia* CopN that has been shown to cause cell cycle arrest in G2/M phase by inhibiting tubulin polymerization (40–42).

In this study, we addressed the mechanism underlying the secretion and translocation activity of T3SS in bordetellae. Specifically, we determined the structure of the BopN protein and investigated its role as a gatekeeper of the T3SS in *B. bronchiseptica*.

## RESULTS

**BopN adopts a conserved structure of the type III secretion gatekeepers.** To gain insight into BopN function in the *Bordetella* injectisome, we first determined its structure by X-ray crystallography and identified structures with a similar fold. To obtain a stable protein fragment of BopN suitable for crystallization experiments, we performed limited proteolysis of the full-length BopN protein (BopN 1 to 365) of *B. pertussis* Tohama I by trypsin. A stable protein fragment with an intact mass of 31,869 Da was obtained corresponding to a truncated BopN protein lacking the first 68 residues (BopN 69 to 365). The protein crystallized in a tetragonal ($P4_2$22) space group with a single molecule per asymmetric unit and its structure was determined with a 1.95 Å resolution (Table 1). As shown by a ribbon representation (Fig. 1A) and by a topology diagram (Fig. 1B), BopN is an elongated protein consisting of three X-bundle domains. The N-terminal domain comprises 5 helices 1 to 5 and includes residues 83 to 161. Residues 69 to 82 are not visible, probably due to conformational flexibility or partial degradation. A kinked helix 6 (residues 162 to 180) connects the N-terminal domain to a central domain consisting of parallel helices 7 and 8 located on top of helices 6, 9, and 10. The C-terminal domain consists of parallel helices 11 and 12, overlying helices 10, 13, and 14. It is connected to the central domain by a shared rigid central helix 10 (residues 243 to 274) and a region (residues 275 to 281) that lacks density in the electron map, indicating loop flexibility.

Structural homology search using the algorithm DALI (43) revealed that the characterized T3SS gatekeepers, namely, *Shigella* MxiC (44), *Chlamydia* CopN (42), EPEC SepL (45), and the *Yersinia* YopN-TyeA complex (46), scored highest (Table 2). Similar to BopN, these proteins exhibit a well-conserved architecture with three four-helix X-bundle domains. However, in the case of YopN, the three-X-bundle domain structure is achieved only after the binding of the TyeA protein to a C-terminal region of YopN (46). The relative orientation of the X-bundle domains in different molecules slightly varies, yielding different overall molecular shapes. When the BopN molecule is superimposed on the structurally characterized gatekeepers (see Fig. S1), the overall shapes and orientations of the X-bundle domains also differ, with the most noticeable difference being the realignment of the N-terminal domain of BopN. This is also reflected in the DALI algorithm Z-scores for BopN (Table 2), which are in good agreement with previously reported Z-scores within the SctW family (45). These are indicative of significant similarity but are typically not high enough to define a strong match between the structures (Z-score > 24 would define a strong match for BopN, (47)).

In summary, BopN structure resembles the structures of other gatekeepers, suggesting that BopN may regulate translocation activity in *Bordetella*, similar to the SctW members of the T3SS injectisomes of other bacteria.

**Inactivation of BopN partially deregulates type III secretion in calcium-rich medium.** It was next important to determine whether BopN serves as the T3SS gatekeeper in *Bordetella* and prevents premature secretion of BteA under secretion-restrictive conditions. Since BopN protein sequences of *B. pertussis* Tohama I and *B. bronchiseptica* RB50 exhibit only a single amino acid difference, namely, a P112T substitution in the loop connecting helices 3 and 4, we chose to perform the subsequent experiments with *B. bronchiseptica*, for which the importance of T3SS function in host respiratory tract infections has been demonstrated (5–7). Moreover, *B. bronchiseptica* has the

**TABLE 1** X-ray diffraction data and collection statistics[a]

| Data processing statistics | |
|---|---|
| Diffraction source | MX 14.1, BESSY II |
| Wavelength (Å) | 0.9184 |
| Temp (K) | 100 |
| Detector | PILATUS3 S 2M |
| Crystal-detector distance (mm) | 297.07 |
| Rotation range per image (°) | 0.1 |
| Total rotation range (°) | 180 |
| Exposure time per image (s) | 0.6 |
| Space group | $P4_2 22$ (93) |
| Cell dimensions | |
| a, b, c (Å) | 84.86, 84.86, 102.57 |
| A, $\beta$, $\gamma$ (°) | 90.0, 90.0, 90.0 |
| Resolution (Å) | 43.93-1.95 (2.00-1.95) |
| Redundancy | 8.88 |
| No. of reflections | |
| Total | 236,216 (36,301) |
| Unique | 26,601 (4,088) |
| Completness (%) | 99.91 (99.85) |
| $R_{meas}$ (%) | 10.4 (12.7) |
| I/$\sigma$ | 18.7 (1.75) |
| CC half (%) | 99 (73) |
| Overall B factor from Wilson plot (Å$^2$) | 36.5 |
| | |
| Refinement statistics | |
| Resolution range (Å) | 43.93–1.95 |
| No. of reflections, working set | 26586 |
| R value[b] (%)/$R_{free}$[c] (%) | 17.97/20.71 |
| No. of atoms: | |
| Protein | 2206 |
| Water | 197 |
| Glycerol | 1 |
| PEG | 1 |
| No. of ions: | |
| Sodium/Calcium ions | 1/1 |
| Acetate ion | 1 |
| Root mean square deviation | |
| Bond lengths (Å) | 0.017 |
| Bond angles (°) | 1.878 |
| Avg B factors (Å$^2$) Overall | 46.01 |
| Ramachandran plot | |
| Most favored region (%) | 98.7 |
| Allowed region (%) | 100.0 |
| PDB code | 7YYG |

[a]The data in parentheses refer to the highest-resolution shell.
[b]R value = $\|F_o| - |F_c\|/|F_o|$, where $F_o$ and $F_c$ are the observed and calculated structure factors, respectively.
[c]$R_{free}$ is equivalent to R value but is calculated for 5% of the reflections chosen at random and omitted from the refinement process.

T3SS constitutively active in the low calcium concentration (0.1 mM Ca$^{2+}$) SS medium and its secreted BteA effector is readily detectable in culture supernatants (9). We further hypothesized that as for other T3SS, a low concentration of calcium ions might trigger effector secretion by *B. bronchiseptica* (24–26), and a millimolar concentration of calcium ions in the growth medium would create secretion-restrictive conditions and prevent BteA secretion.

To quantitatively monitor the BteA content in bacterial cells and the secreted fractions, we implemented a split-luciferase system (48–50) and used a nontoxic *bteA*[rep] reporter strain of *B. bronchiseptica* RB50 in which the N-terminal 130-amino acid segment of BteA was tagged at the C-terminus with a HiBiT peptide. As shown schematically in Fig. 2A, the split-luciferase reporter system exploits a high-affinity functional complementation between a small 11-amino acid tag called HiBiT and a larger 18-kDa fragment LgBit. The complex exerts a luciferase activity and emits light in the presence of

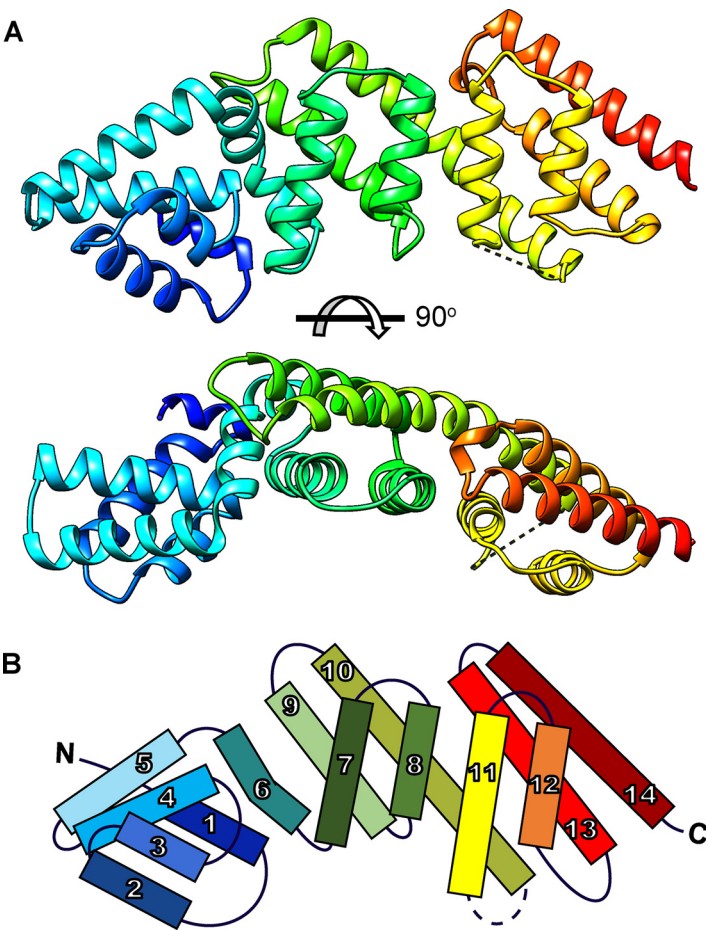

**FIG 1** Structure of the BopN protein. (A) Ribbon representation of BopN residues 83 to 365. The bottom structure is a 90° rotation around the horizontal axis of the BopN structure shown above. The broken line between BopN residues 276 and 282 denotes the disordered region of the structure. This figure was generated by Chimera 1.14rc. (B) Topology diagram of the BopN protein. The helices 1 to 14 are arranged in 3 X-bundle domains. BopN domains 1 and 2 are connected by a kinked helix 6, whereas domains 2 and 3 are connected by the rigid central helix 10 and a region with no density in the electron map.

added furimazine substrate. Therefore, the amounts of BteA^rep in the intracellular and secreted fractions can be easily quantified upon the addition of the LgBit fragment and furimazine. Indeed, as shown by dilution series of purified recombinant BteA^rep protein in the presence of LgBit fragment in excess, the generated luminescence signal is linear over several orders of magnitude of BteA^rep concentrations and is not affected by the concentration of calcium ions (Fig. 2B). Importantly, as shown in Fig. 2C, analysis of BteA^rep amounts in late exponential cultures of *bteA*^rep revealed that growth in calcium-rich (2 mM) medium decreased the secreted amounts of BteA^rep without affecting its intracellular levels, compared to medium without calcium ions. Indeed, BteA^rep was

**TABLE 2** Proteins structurally related to BopN[a]

| Protein | PDB/chain | z-score | RMSD (Å) | No. of Cα atoms aligned | No. of residues | % id of aligned residues |
|---|---|---|---|---|---|---|
| *Shigella* MxiC | 2vj5-B | 20.9 | 4.5 | 265 | 283 | 17 |
| *Chlamydia* CopN | 6gx7-H | 20.2 | 7.3 | 264 | 290 | 17 |
| *E. coli* SepL | 5c9e-B | 14.1 | 4.0 | 182 | 263 | 15 |
| *Yersinia* TyeA | 1xl3-D | 12.8 | 1.7 | 83 | 85 | 23 |
| *Yersinia* YopN | 1xl3-B | 12.7 | 3.0 | 175 | 206 | 19 |

[a]Structure of the BopN protein was correlated using DALI (distance matrix alignment) algorithm (43). The highest-ranking structures are shown.

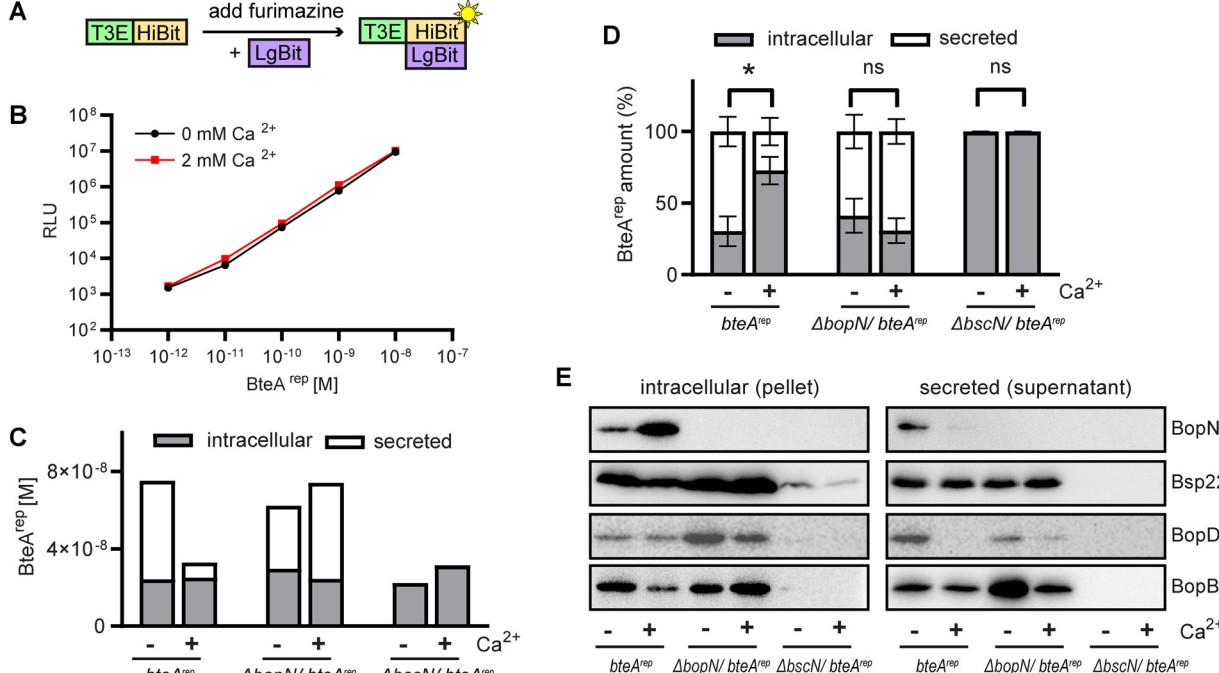

**FIG 2** Inactivation of BopN leads to a partial deregulation of the type III secretion in calcium-rich medium. (A) Schematic representation of HiBiT-LgBit functional complementation for quantification of protein amounts. The HiBit peptide fused to the protein of interest, e.g., type III effector, T3E, binds to added LgBit. The complex generates a luminescent signal in the presence of furimazine substrate. (B) Linearity of luminescence generated by BteA$^{rep}$-LgBit complementation. Luminescence was measured after the addition of LgBit and furimazine substrate to the indicated amounts of purified recombinant BteA$^{rep}$ in the absence or presence of 2 mM Ca$^{2+}$. Shown are mean values ± SD of triplicate wells from a representative experiment of 3 experiments performed. (C) and (D) BopN controls secretion of BteA. The amounts of intracellular and secreted BteA$^{rep}$ in exponential cultures grown in the absence or presence of 2 mM Ca$^{2+}$ were determined by luminescence measurements after appropriate dilution. In (C), the absolute amounts of BteA$^{rep}$ in fractions of the representative culture are shown. In (D), the amount of BteA$^{rep}$ in the fraction was expressed as % of total BteA$^{rep}$ in the culture. Values represent the means ± SD of 3 independent experiments. Statistical analysis was performed using an unpaired two-tailed *t* test; * $P < 0.05$; ns, not significant. (E) BopN does not regulate secretion of the tip protein Bsp22 and translocators BopD and BopB. Supernatants and pellets from overnight cultures of the b*teA*$^{rep}$ strain and its derivatives were analyzed by immunoblotting with anti-BopN (BopN, 1:10,0000), anti-Bsp22 (Bsp22, 1:10,000), anti-BopD (BopD, 1:10,0000), and anti-BopB (BopB, 1:10,000) antisera. Results are representative of 3 independent experiments.

secreted rather inefficiently in calcium-rich medium, with a BteA$_{out}$/BteA$_{in}$ ratio of ~ 0.3. In contrast, in the absence of calcium ions, the ratio of BteA$_{out}$/BteA$_{in}$ was ~ 2.3, as further shown in Fig. 2D. Hence, in the presence of 2 mM calcium ions, BteA secretion decreased 7-fold, suggesting that calcium-rich media creates secretion-restrictive conditions. As expected, negligible amounts of BteA$^{rep}$ were detected in the culture medium of the type III secretion-deficient Δ*bscN*/*bteA*$^{rep}$ mutant (deleted BscN ATPase essential for T3SS function), showing that BteA$^{rep}$ excretion into supernatants specifically depends on T3SS functionality. Importantly, upon *bopN* gene deletion from the genome of the reporter strain, BteA$^{rep}$ secretion was not blocked in the calcium-rich medium anymore (Fig. 2C and D). Indeed, the Δ*bopN*/*bteA*$^{rep}$ strain culture yielded a BteA$_{out}$/BteA$_{in}$ ratio of ~ 2.3 even in calcium-rich medium, whereas this ratio was ~ 1.3 in the absence of calcium ions. Moreover, consistent data were obtained at all phases of the *B. bronchiseptica* growth curve (early exponential, late exponential, and stationary growth), as shown in Fig. S2, demonstrating that BopN prevents secretion of BteA under secretion-restrictive conditions.

We next characterized the secretion of other *Bordetella* T3SS components under secretion-restrictive (2 mM Ca$^{2+}$) and secretion-permissive (0 mM Ca$^{2+}$) conditions, focusing on the role of the BopN protein in secretion of the T3SS tip protein Bsp22 and of the translocator proteins BopD and BopB. Interestingly, immunoblot analysis of culture supernatants and pellets from overnight (early stationary) cultures revealed that compared to secretion-permissive conditions, also BopN secretion was prevented in secretion-restrictive medium (Fig. 2E and Fig. S3). In contrast, no significant impact on

secretion of the tip protein Bsp22 and of the translocator proteins BopB and BopD was observed in secretion-restrictive medium or upon inactivation of the *bopN* gene. Whereas the trends of BopN and Bsp22 detection in intracellular and secreted fractions were consistent and highly reproducible, a high variation in detected amounts of BopB and BopD proteins between fractions was observed for unclear reasons (Fig. 2E and Fig. S3).

In summary, hence, BopN exerted a control over the secretion of the BteA effector but not over the secretion of the Bsp22, BopB and BopD components of the injectisome. Inactivation of the *bopN* gene then led to enhanced secretion of BteA in secretion-restrictive culture conditions (2 mM $Ca^{2+}$). Furthermore, data also indicated that secretion of the BopN protein itself was prevented under secretion-restrictive conditions.

**BopN responds to low calcium ion concentration and localizes to *Bordetella* cell periphery in calcium-rich medium.** To corroborate the mechanism of BopN action, we next constructed a *bopN^rep* reporter strain of *B. bronchiseptica* RB50 expressing a chromosomally-encoded BopN^rep protein tagged at its C-terminus with the peptide HiBiT-3xFLAG-SPOT. The resulting bopN^rep strain exhibited the same T3SS-dependent cytotoxicity against A549 lung epithelial cells as the wild-type strain (Fig. S4A), demonstrating that the tagging of BopN did not affect T3SS injectisome function. As in the case of recombinant BteA^rep, dilution series of purified recombinant BopN^rep to which LgBit was added in excess showed that binding of BopN^rep to LgBit produced a luminescence signal that was linear over several orders of magnitude of the BopN concentrations and was independent of calcium ions (Fig. 3A), thus allowing quantitative monitoring of BopN^rep secretion in response to calcium ions. To this end, *bopN^rep* bacteria grown in calcium-rich medium (2 mM $Ca^{2+}$) were transferred into medium with various $Ca^{2+}$ concentrations (0, 0.1, 0.5, 1, and 2 mM) and incubated for 90 min. Subsequently, the amounts of secreted and intracellular BopN^rep protein were determined by luminescence measurements and further verified by immunoblot analysis. As depicted in Fig. 3B and C, and Fig. S4B, in the absence of calcium ions in the medium about 50% of the BopN^rep molecules were secreted and ~ 50% remained intracellular. A slightly lower secretion efficiency of BopN^rep was observed in a medium with 0.1 mM $Ca^{2+}$. In contrast, at high $Ca^{2+}$ concentrations of 0.5, 1, and 2 mM, the majority of produced BopN^rep molecules were intracellular, indicating a very low secretion efficacy. Interestingly, as also shown in Fig. 3B, deletion of the *bscN* gene encoding for the T3SS ATPase essential for T3SS function, yielded a decrease of intracellular BopN^rep level compared to the wild-type strain, likely due to a negative feedback loop on BopN^rep transcription/translation and/or stability. Overall, these data demonstrated that BopN secretion is prevented in the secretion-restrictive calcium-rich medium.

We next examined the localization of BopN^rep in cells of *B. bronchiseptica* using structured illumination microscopy (SIM). As revealed by immunofluorescence labeling with an anti-SPOT nanobody conjugated to ATTO594 (Fig. 3D), the BopN^rep protein localized to the bacterial periphery under secretion-restrictive conditions (2 mM $Ca^{2+}$). The majority of BopN^rep foci were located beneath the bacterial cell envelope and were more concentrated at the poles of the bacterial cells. In contrast, under secretion-permissive conditions (0 mM $Ca^{2+}$) much less intracellular BopN^rep foci were detected and their preferential association with cell poles was lost (Fig. 3D and Fig. S4C). To determine whether the distribution of the BopN^rep foci under the calcium-rich conditions was consistent with the localization of *Bordetella* injectisomes, we tagged the needle complex inner ring component BscD at its cytoplasmic N-terminus with the mNeonGreen (mNG) fluorescent protein. Importantly, tagging of the BscD protein did not affect injectisome function, as the derived *mNG-bscD/bopN^rep* strain elicited the same T3SS-dependent cytotoxicity as the wild-type or *bopN^rep* reporter strain (Fig. S4A). Furthermore, as shown in Fig. 4, the mNG-BscD fusion protein localized to the cell periphery, like BopN^rep. Interestingly, only a partial overlap of localization of BopN^rep and mNG-BscD proteins was observed. The mNG-BscD foci were more numerous than the BopN^rep foci and exhibited a more regular pattern at the cell periphery (Fig. 4). In summary, these results show that under secretion-restrictive culture conditions

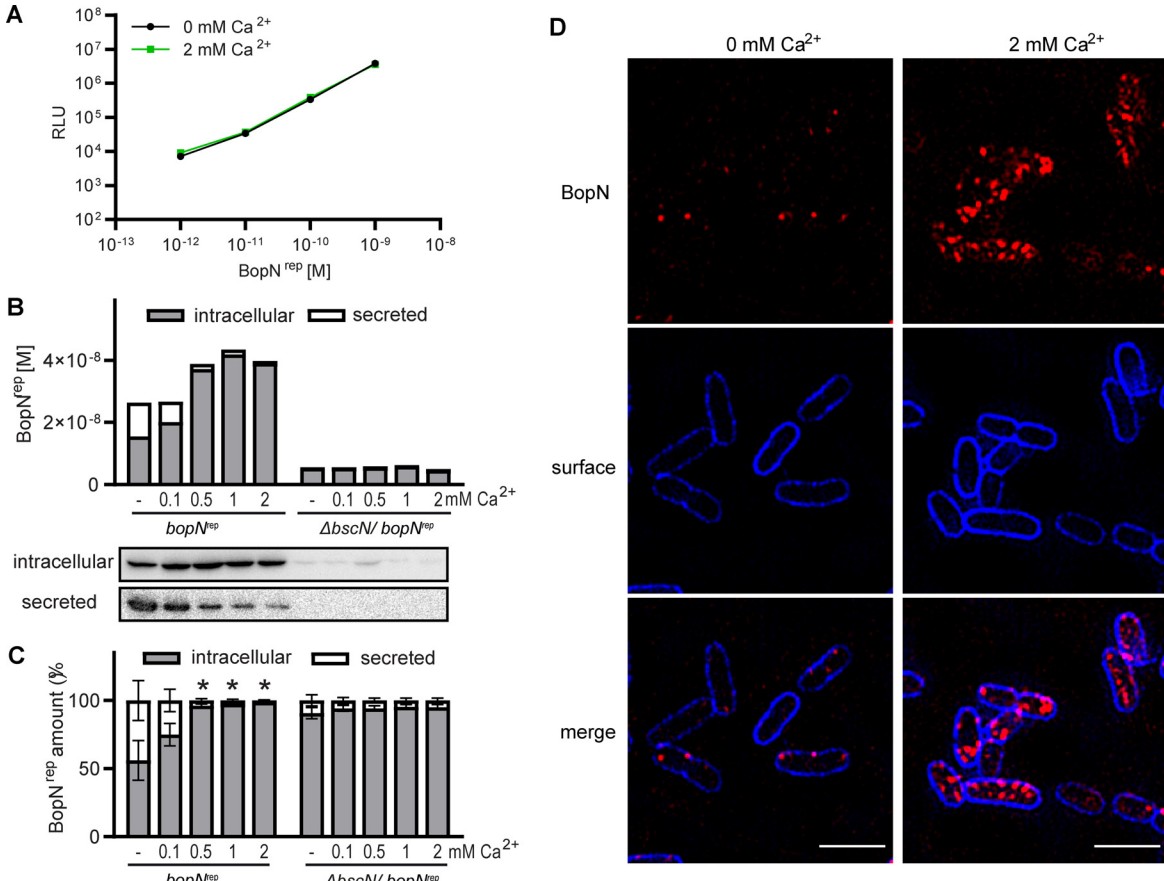

**FIG 3** BopN responds to low Ca²⁺ concentration and localizes to the cell periphery. (A) Linearity of luminescence generated by BopN$^{rep}$-LgBit complementation. Luminescence was measured after the addition of LgBit and furimazine substrate to the indicated amounts of purified recombinant BopN$^{rep}$ in the presence or absence of 2 mM Ca²⁺. Shown are mean values ± SD of triplicate wells from a representative experiment of 3 experiments performed. (B) and (C) Calcium-rich medium prevents BopN secretion. Cells of the *bopN$^{rep}$* and the secretion-deficient Δ*bscN*/*bopN$^{rep}$* derivative were incubated at the indicated concentration of Ca²⁺ for 90 min. The amounts of intracellular and secreted BopN$^{rep}$ were determined by luminescence measurements after appropriate dilution. In (B) the absolute amounts and immunoblot detection of BopN$^{rep}$ in fractions of the representative culture are shown. In (C), the amount of Bop$^{rep}$ in the fraction was expressed as % of total BopN$^{rep}$ in the culture. Values represent the mean ± SD of 3 independent experiments. Significant differences (*, $P < 0.05$, unpaired two-tailed *t* test) between BopN$^{rep}$ secreted in the absence of calcium and the corresponding culture at the indicated calcium ion concentration are marked. (D) BopN$^{rep}$ localizes to the periphery of *Bordetella* cells in calcium-rich medium. Cells of *bopN$^{rep}$* were fixed on polylysine-coated coverslips after incubation for 90 min in the absence or presence of 2 mM Ca²⁺. BopN$^{rep}$ was visualized with an anti-SPOT nanobody conjugated to ATTO594, whereas the outer surface of the bacteria was stained with a rabbit anti-*Bordetella* serum and detected with the DyLight 405-labeled anti-rabbit IgG conjugate. A single focal plane of a Z-stack is shown. Fluorescence images are representative of 3 independent experiments. Scale bar represents 2 $\mu$m.

(2 mM Ca²⁺), BopN is localized to the periphery of *Bordetella* cells and is exported in response to a low calcium concentration, which is an artificial trigger of type III secretion in *B. bronchiseptica*.

**BopN is injected early upon contact with host cells but does not modulate NF-$\kappa$B signaling in epithelial cells.** To gain further insight into the mode of action of BopN, we next examined its fate upon bacterial contact with host cells. Specifically, we determined the translocation kinetics of BopN$^{rep}$ into host cells using the split-luciferase system. Toward this aim, we generated a nontoxic *bopN$^{rep}$*/Δ*bteA B. bronchiseptica* strain by in-frame deletion of the *bteA* open reading frame and expressed LgBit in the host cell. In this scenario, shown schematically in Fig. 5A, LgBit can be complemented only by the HiBit-tagged BopN$^{rep}$ protein injected into the cell cytosol by the T3SS. In the presence of a cell-permeable furimazine substrate, the formed LgBit/HiBit-tagged BopN$^{rep}$ complex then generates a luminescence signal (48, 49).

Accordingly, as shown in Fig. 5B, no luminescence signal was detected when LgBit-expressing HeLa cells were infected with a Δ*bsp22*-derivative containing a deletion of

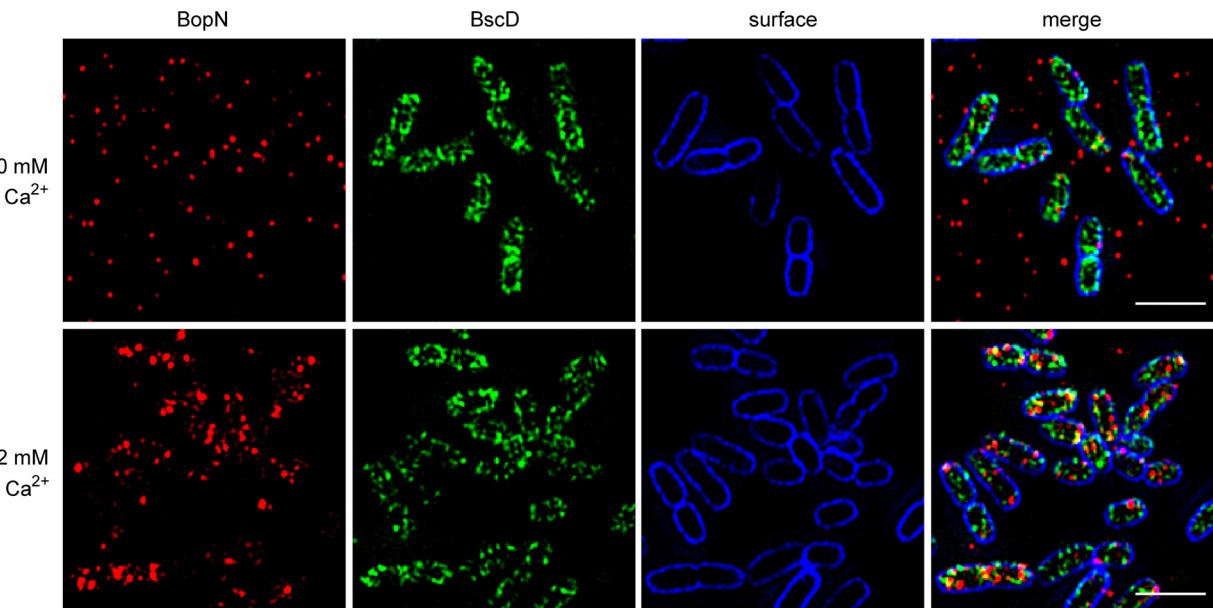

|  | BopN | BscD | surface | merge |

**FIG 4** Localizations of BopN protein and the injectisome inner ring component mNG-BscD partially overlap in calcium-rich medium. Cells of *mNG-bscD/bopN*rep were fixed on polylysine-coated coverslips after incubation for 90 min in the presence or absence of 2 mM Ca²⁺. BopNrep was visualized with an anti-SPOT nanobody conjugated to ATTO594, while the outer surface of *Bordetella* was stained with a rabbit anti-*Bordetella* serum followed by anti-rabbit IgG-DyLight 405 conjugate. A single focal plane of a Z-stack is shown. Fluorescence images shown are representative of 3 independent experiments. Scale bar represents 2 μm.

the open reading frame for the T3SS tip protein (Δ*bsp22/bopN*rep/Δ*bteA*) or a secretion-deficient Δ*bscN* derivative (Δ*bsN/bopN*rep/Δ*bteA*) unable of T3SS-mediated injection. In contrast, a steady increase in luminescence was observed for up to 60 min postinfection when LgBit-expressing HeLa cells were infected with the *bopN*rep/Δ*bteA* strain at the same multiplicity of infection (MOI) 5:1 (Fig. 5B). These data indicate that a preexisting pool of BopNrep molecules was injected early upon bacterial contact with host cells.

We also investigated whether BopN-injected into host epithelial cells modulates their NF-κB signaling. To this end, human A549 alveolar basal reporter cells, encoding the secreted embryonic alkaline phosphatase (SEAP) under the control of the NF-κB-responsive promoter, were infected with *B. bronchiseptica* bacteria and SEAP activity was determined in the culture supernatants. We were unable to perform these experiments with wild-type *B. bronchiseptica* due to acute BteA-induced cytotoxicity and cell lysis. The BteA-secreting wild-type or Δ*bopN* strains of *B. bronchiseptica* did not elicit any detectable SEAP production at MOI 1:1, 5:1 or 25:1 (data not shown). In contrast, SEAP activity in the supernatant increased with increasing MOI when cells were infected with Δ*bteA* and Δ*bopN*/Δ*bteA* derivatives (Fig. 5C). Importantly, no statistically significant difference (unpaired two-tailed *t* test using the corresponding MOI) was observed between the SEAP activities in the supernatants from Δ*bteA* and Δ*bopN*/Δ*bteA* bacteria-infected cells (Fig. 5C). Overall, these data demonstrate that BopN is injected early upon contact with host cells but does not modulate the NF-κB-signaling in epithelial cells.

**BopN is required for efficient and targeted BteA injection into host cells.** Having shown that BopN controls BteA secretion and is itself injected into host cells early upon bacterial contact with cells, it was important to investigate whether deletion of the *bopN* gene will also result in the loss of control over BteA delivery to host cells. To this end, we analyzed the amounts of BteArep that leaked into the culture medium and the amounts of BteArep that were injected into host cells during infection.

Leakage of BteArep into the medium during infection of HeLa cells was quantified as luminescence signal after addition of LgBit protein and furimazine substrate to aliquots

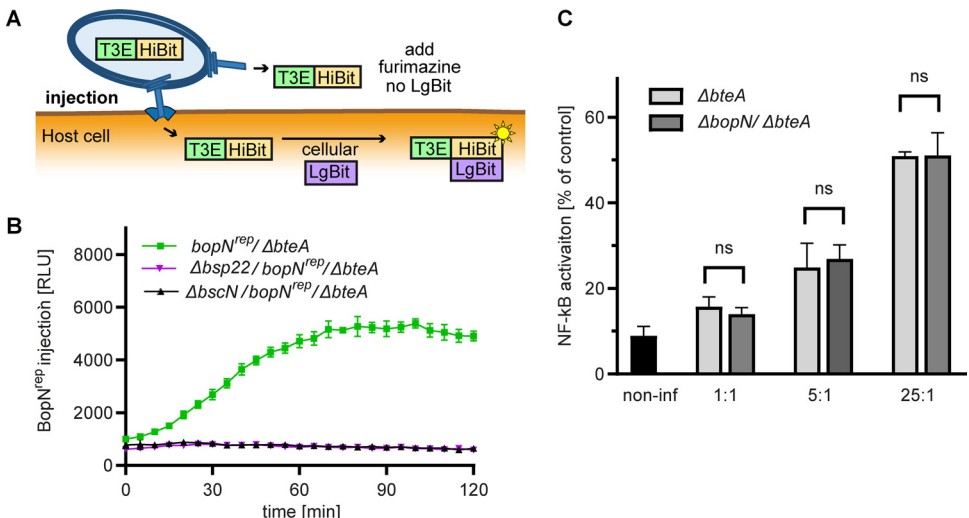

FIG 5 BopN is injected early upon contact with host cells and does not on its own modulate NF-κB signaling in epithelial cells. (A) Schematic representation of HiBit-LgBit functional complementation for real-time monitoring of protein injection. LgBit-expressing host cells are infected in the presence of a cell-permeable furimazine substrate. Injection of the HiBit-fused protein of interest, T3E, is detected as a luminescent signal after complementation with LgBit expressed in the host cell cytosol. The absence of LgBit in the medium surrounding the cells prevents the detection of any amounts of the secreted T3E. (B) BopN is injected early upon contact with host cells. LgBit-expressing HeLa cells were infected with *bopN*rep/Δ*bteA* and the indicated derivates at MOI of 5:1. Luminescence measurements were performed at 5-min intervals and were expressed as relative luminescence units (RLU). Shown are mean values ± SD of triplicate wells from a representative experiment of 3 experiments performed. (C) BopN does not modulate NF-κB signaling in epithelial cells. A549 Dual reporter cells encoding secreted embryonic alkaline phosphatase (SEAP) under the control of the NF-κB-responsive promoter were infected at the indicated MOI. The amount of secreted SEAP in the culture medium was determined after 20 h of infection and is expressed as % of SEAP in the culture medium of cells stimulated with 1 ng/mL TNF-α. Values represent the means ± SD from 2 independent experiments (*n* = 4). Statistical analysis was performed using an unpaired two-tailed *t* test; ns, not significant.

of the spent medium. Interestingly, the Δ*bopN* (Δ*bopN*/*bteA*rep) bacteria leaked approximately 10-fold higher amounts of BteArep into culture medium than the wild-type reporter *bteA*rep *B. bronchiseptica* strain (Fig. 6A). In contrast, such increase was not observed with the T3SS-tip-deficient derivative (Δ*bsp22*/*bteA*rep), which exhibited a lower leakage of BteArep than the wild-type strain. As expected, negligible amounts of leaked BteArep were detected in infection experiments with the secretion-deficient strain (Δ*bscN*/*bteA*rep) (Fig. 6A), confirming that occurrence of the luminescence signal depended on T3SS function. When the same reporter strains were used to quantify the injection of BteArep into LgBit-expressing HeLa cells by luminescence measurements, the wild-type *bteA*rep strain translocated the largest amounts of BteArep into the cells (Fig. 6B). The luminescence observed for the Δ*bopN* strain (Δ*bopN*/*bteA*rep), reflecting the amount of BteArep injected into the cells, was reproducibly lower and slightly delayed. As expected, no injection of BteArep was detected with the translocation-deficient (Δ*bsp22*/*bteA*rep) and the secretion-deficient (Δ*bscN*/*bteA*rep) strains, respectively (Fig. 6B). Overall, these data suggest that the BopN protein is required for effective and targeted delivery of BteA by *Bordetella* T3SS into host cells.

To confirm these results and indirectly evaluate the efficiency of T3SS-mediated injection of the cytotoxic effector BteA into host cells, we assessed the BteA-induced cytotoxicity toward human A549 alveolar basal epithelial cells and murine Raw 264.7 macrophages. Consistent with previous results, infection with the Δ*bopN* mutant yielded lower cytotoxicity on A549 cells and Raw 264.7 macrophages than infection with the wild-type bacteria (Fig. 6C and D). This confirmed the defect in T3SS-mediated BteA injection into cells in the absence of the BopN protein. Indeed, the observed cytotoxicity of *B. bronchiseptica* toward host cells was due to the action of the effector BteA, as the Δ*bteA* mutant strain did not cause any early cytotoxicity at the MOIs used for A549 cells (5:1) or Raw 264.7 macrophages (1:1).

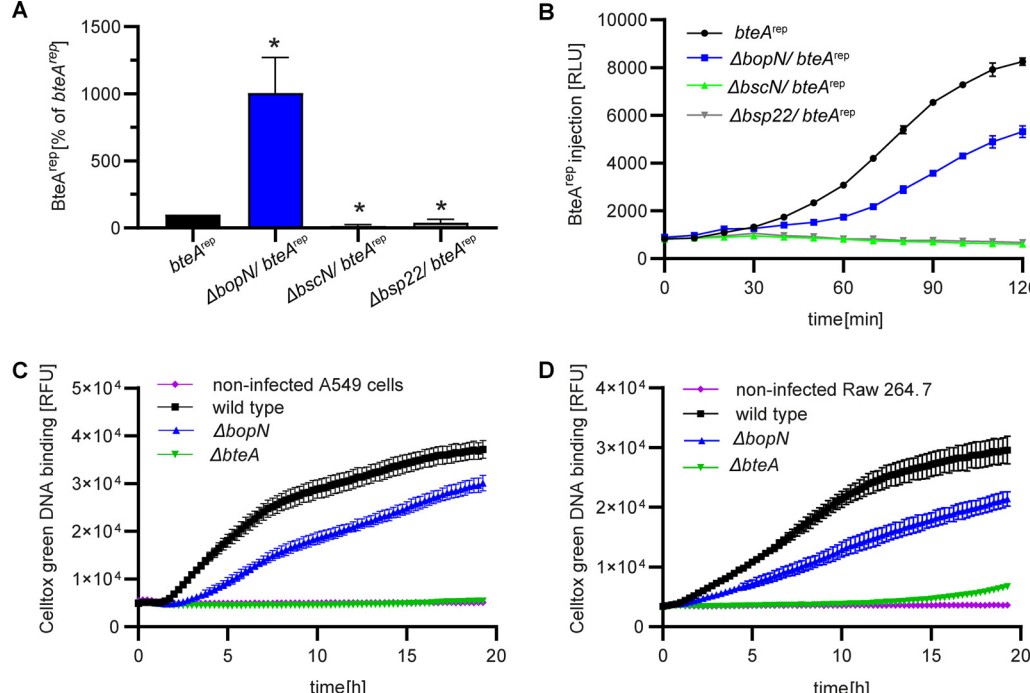

**FIG 6** BopN is required for efficient and targeted BteA injection into host cells. (A) BopN controls the leakage of BteA into the surroundings of host cells during infection. HeLa cells were infected with the *bteA*rep and the indicated derivatives at MOI of 5:1. The amount of BteA rep in the medium surrounding the cells was determined by luminescence measurements after 3 h of infection and expressed as % of BteA rep detected in the medium of *bteA*rep-infected cells. Values represent the means ± SD from 3 independent experiments (*n* = 3). Significant differences (*, $P < 0.05$, unpaired two-tailed *t* test) between BteA rep secreted by derivative and *bteA*rep strains are indicated. (B) Inactivation of BopN leads to defects in the delivery of BteA into host cells. LgBit-expressing HeLa cells were infected with *bteA*rep and the indicated derivates at MOI of 5:1. Luminescence measurements were performed at 5-min intervals and are reported as relative luminescence units (RLU). Shown are mean values ± SD of triplicate wells from a representative experiment. Data are representative of 3 independent experiments. (C) and (D) Inactivation of BopN results in reduced BteA-mediated cytotoxicity. Toxicity of the indicated strains against human A549 alveolar epithelial cells infected at MOI 5:1 (C) and against mouse Raw 264.7 macrophages infected at MOI 1:1 (D) was measured as real-time membrane permeabilization kinetics and monitored by fluorescence of the DNA-binding dye CellTox Green. Shown are mean values ± SD of triplicate wells from a representative experiment. Data are representative of 2 independent experiments.

In summary, our data show that BopN controls the targeted delivery of BteA into the host cell cytosol and that BopN is the gatekeeper of T3SS in bordetellae.

## DISCUSSION

In this study, we have solved the structure of the BopN protein fragment (residues 83 to 365) at a refinement 1.95 Å and report that it is similar to the structures of previously characterized T3SS gatekeeper proteins of Gram-negative bacteria. We further demonstrate experimentally that BopN is a gatekeeper of the *Bordetella* T3SS.

The T3SS is employed by many bacterial pathogens to translocate effector proteins directly into host cell cytosol to modulate host cell functions to the advantage of the bacteria. While translocated effector proteins and the transcriptional and posttranscriptional network regulating the T3SS expression are unique to each species, the biogenesis of T3SS injectisomes, their structural components and their overall architecture are broadly conserved (51). Biogenesis of the injectisome is a tightly regulated hierarchical process in which the T3SS substrates are secreted in a specific order (22, 23). Effector proteins are secreted only when the T3SS needle tip senses the host cell *in vivo* or in the presence of an artificial trigger (25, 27–29). Here, we demonstrated that the T3SS secretion of *B. bronchiseptica* is activated by a low concentration of calcium ions, similar to T3SS secretion of *Yersinia* and EPEC (24–26). The standard SS medium used to culture *Bordetella* cells contains a low calcium ion concentration (~ 0.1 mM) and, therefore,

artificially activates *Bordetella* T3SS secretion. When the *Bordetella* culture medium contains calcium ions at the 2 mM physiological concentration found in body fluids, secretion of the BteA effector is inhibited ($\sim$ 7-fold). However, how the artificial signal or host cell contact is transmitted to trigger BteA secretion remains to be elucidated, and both mechanical and chemical signals need to be considered (52).

One protein that is critical for triggering the secretion of effectors and their targeted delivery into host cells is the T3SS gatekeeper, referred to as SctW. It has been suggested that the gatekeeper interacts with the SctV protein of the T3SS export apparatus, where it supports the loading of translocator-chaperone complexes but blocks the entry of effector-chaperone complexes. Following an activation signal (contact with the host cell and/or an artificial trigger), the gatekeeper is removed to allow the secretion of effector proteins (31, 33).

Although BopN shares a sequence similarity with YopN and other T3SS gatekeepers (Table 2), its function as a gatekeeper in *Bordetella* injectisome has been controversial. First, the deletion of the *bopN* gene does not affect the overall amounts of T3SS-secreted proteins released into the standard low calcium-containing SS medium (10). Second, the inactivation of BopN was reported to impair BteA-mediated cytotoxicity toward rat L2 pulmonary epithelial cells but not toward murine DC2.4 dendritic cells (10, 19). Third, BopN was found to be translocated into host cells and was reported to impair *B. bronchiseptica*-induced NF-$\kappa$B signaling (10, 19). Importantly, we show here that the absence of BopN promotes secretion of the BteA effector under the secretion-restrictive conditions when *B. bronchiseptica* is grown in calcium-rich modified Stainer-Scholte (SSM) medium, which reveals the gatekeeper function of BopN. Further, we also demonstrated that *B. bronchiseptica* bacteria lacking BopN exhibit approximately 10-fold increased leakage of BteA into the culture medium during cell infection experiments and impair BteA injection into host cells, which yields reduced BteA-mediated cytotoxicity. Hence, although BopN is required for targeted and efficient translocation of BteA into host cells, its inactivation does not entirely prevent the delivery of BteA into host cells. This is most likely because the absence of BopN does not hinder the secretion of the translocators BopB and BopD and the tip filament Bsp22. Thus, like for the YopN/TyeA gatekeeper complex of *Yersinia* spp. (53, 54), BopN is not required for loading the translocator-chaperone complexes into the export apparatus of *Bordetella* injectisome. Indeed, the gatekeeper-mediated control of translocator secretion is more complex and can differ among species. Most gatekeeper mutants, e.g., mutants of InvE of the *Salmonella* pathogenicity island 1 (SPI-1) (55) and SsaL of the SPI-2 (27, 33), SepL of EPEC (26), VgpA, and VgpB of *Vibrio* (29) exhibit defects in their secretion, and thereby efficiently block effector translocation into host cells. In contrast, the MxiC mutant of *Shigella* does not impair the translocator protein secretion (38) and the YopN/TyeA mutant of *Yersinia* even upregulates their secretion (53, 54).

The here presented functional data are also supported by the solved structure of truncated BopN (residues 83 to 365) that exhibits structural similarity to the characterized gatekeeper proteins belonging to the YopN/InvE/MxiC (or LcrE) family (InterPro entry IPR013401). The BopN structure is rod-shaped and consists of 3 X-bundle domains with 4 $\alpha$-helices, potentially allowing simultaneous interactions with multiple partner proteins, while having a disordered N-terminus containing the secretion signal and a chaperone-binding region not included in the structure (45, 56). Interestingly, the BopN molecule exhibits a different relative orientation of the 3 X-bundle domains compared to other gatekeepers (Fig. S1), which could reflect the contribution of the missing N-terminal part to the overall structure, a different state of the trapped BopN molecule due to different crystallization conditions, or an actual difference of the BopN molecule. At present, it is not known why some gatekeepers, such as BopN or YopN, are secreted by the injectisome upon activation signal, whereas others, such as SepL, are not secreted. The presence of a secretion signal *per se* does not provide an explanation, as this was revealed also in the non-secreted SepL protein upon truncation of its C-terminal portion (57). Thus, the fate of the gatekeeper more likely depends on its

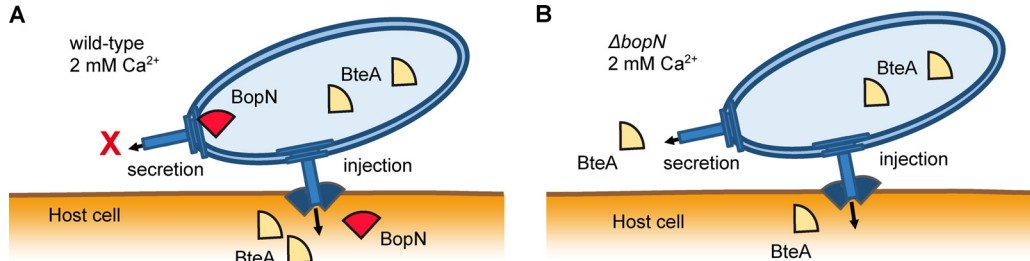

**FIG 7** Proposed model for the BopN action in bordetellae. (A) In wild-type cells, BopN prevents secretion of BteA before contact with the host cell. The host cell contact with the injectisome triggers translocation of BopN into the host cell and also allows for targeted translocation of BteA. (B) In the Δ*bopN* mutant, targeted BteA translocation does not take place due to the absence of BopN, therefore, BteA leaks into the surroundings of the infected cells. This results in lower amounts of BteA being injected into the host cells.

interactions with other components of the injectisome and probably also on the ATPase SctN, which accounts for the ATP-dependent release of chaperones and for the unfolding of the corresponding secreted proteins (58). Indeed, further studies are needed to investigate the interactions of BopN in *Bordetella* T3SS and to uncover the mechanisms of how BopN senses the low calcium concentration and activates the secretion and translocation of BteA. To gain an initial insight into the action of BopN, we analyzed localization of BopN in *B. bronchiseptica* cells and its distribution compared to T3SS injectisomes. We found that BopN distributed in patches beneath the cell surface. The tagged mNG-BscD, inner ring component of the needle complex, also localized in foci at the cell periphery. These foci were visible in all cells and resembled the foci of SctD from *Y. enterocolitica* (59, 60). Interestingly, the BopN foci appeared to be located more in the cytoplasm than the BscD foci and were less numerous. Not all BscD foci were occupied by BopN. The number of intracellular BopN foci decreased under secretion-promoting conditions, confirming that BopN is secreted in response to an artificial trigger. We also demonstrated that BopN is released from bacterial cells and translocated into the host cells early after the host cell contact. However, we did not detect any BopN-mediated modulation of NF-$\kappa$B signaling, contrary to what has been reported previously (19). This may be related to the use of different cells and assays.

Overall, our study provides new insights into the action of the *Bordetella* BopN protein by showing that BopN protein promotes efficient and targeted T3SS-mediated injection of BteA into host cells and acts as the T3SS gatekeeper, as summarized by the model in Fig. 7. These findings are important for understanding the mechanism of action of the T3SS of *Bordetella* and for future deciphering of the role of its effector(s) in *Bordetella* infections.

## MATERIALS AND METHODS

**Bacterial strains and growth conditions.** The bacterial strains used in this study are listed in Table S1. *E. coli* strain XL1-Blue was used for plasmid construction, *E. coli* strain SM10$\lambda$ pir was used for plasmid transfer into *B. bronchiseptica* RB50 by bacterial conjugation, and *E. coli* BL21 $\lambda$(DE3) and XL-1 Blue were employed for expression of recombinant proteins. *E. coli* strains were cultivated at 37°C in LB agar or LB broth. When appropriate, the LB medium was supplemented with 100 $\mu$g/mL of ampicillin or 60 $\mu$g/mL of kanamycin. The parental *B. bronchiseptica* RB50 and derived mutant strains were grown on Bordet-Gengou (BG) agar medium (Difco) supplemented with 1% glycerol and 15% defibrinated sheep blood (LabMediaServis) at 37°C and 5% $CO_2$ for 48 h. Liquid cultures of *Bordetella* strains were performed in modified Stainer-Scholte (SSM) medium (61) supplemented with 5 g/l of Casamino Acids (Difco) at 37°C. To maximize the expression of the type III secretion system for assays, SSM medium was formulated with reduced l-glutamate (monosodium salt) concentration (11.5 mM, 2.14 g/l) and no $FeSO_4.7H_2O$ added (62, 63). To obtain the indicated concentration of calcium ions within SSM, 1 M $CaCl_2$ was used. *B. bronchiseptica* for assays and inoculations was grown to a mid-exponential phase ($OD_{600}$ 1.5) in calcium-rich SSM medium (2 mM $Ca^{2+}$) unless otherwise indicated.

**Cell culture and generation of LgBit-expressing HeLa cells.** Dulbecco's Modified Eagle Medium ([DMEM] Sigma) supplemented with 10% fetal bovine serum (DMEM-10% [FBS]), was used to cultivate the following cell lines at 37°C and 5% $CO_2$: HeLa (ATCC CCL-2, human cervical adenocarcinoma), LgBit-expressing HeLa (see below), 293T (ATCC CRL-3216, human epithelial kidney cell line), A549 (ATCC CCL-185, human lung carcinoma epithelial cells), A549-Dual (InvivoGen a549d-nfis, A549 cell reporter derivatives encoding secreted embryonic alkaline phosphatase [SEAP] under the control of the IFN-$\beta$ minimal

promoter fused to five NF-$\kappa$B binding sites) and RAW 264.7 macrophages (ATCC TIB 71, Abelson leukemia virus-transformed murine cell line). LgBit-expressing HeLa cells were generated by lentiviral transduction of the parental HeLa cell line. The coding sequence of LgBit was subcloned into a modified pLJM1-EGFP vector (Addgene #19319) in-frame with N-terminal FLAG epitope. VSV-pseudotyped viruses were then produced by co-transfection of 6 $\mu$g pLJM1-FLAG-LgBit, 6 $\mu$g pCMV-VSV-G, and 6 $\mu$g psPAX2 plasmids using Lipofectame 2000 (Invitrogen) into 293T cells grown in a 10-cm dish. The cell culture supernatant was collected 48 h after transfection and used to transduce the parental HeLa cells in the presence of Polybrene (8 $\mu$g/mL). Twenty-four hours after transduction, cells were split, and transduced cells were selected by puromycin (0.5 $\mu$g/mL). Expression of flag-LgBit was verified by immunoblotting with the anti-flag M2 antibody (Sigma-Aldrich).

**Plasmid construction and *Bordetella* allelic exchange.** Plasmids used in this study are listed in Table S2, and were constructed using the T4 DNA ligase or Gibson assembly strategy (64). PCR amplifications were performed from chromosomal DNA of *B. pertussis* Tohama I or *B. bronchiseptica* RB50 using Herculase II Phusion DNA polymerase (Agilent). The LgBit coding sequence was synthesized as GeneArt strings fragments (Invitrogen, Thermo Fisher Scientific), and mNeonGreen coding sequence was amplified from mNeonGreen-Rab7 vector (Addgene #129603). All constructs were verified by DNA sequencing (Eurofins Genomics). Mutant *B. bronchiseptica* strains were constructed by homologous recombination using the suicide allelic exchange vector pSS4245 as previously described (18). The presence of the introduced mutations was confirmed by PCR amplification of the relevant regions of the *Bordetella* chromosome, followed by agarose gel analysis and DNA sequencing (Eurofins Genomics).

**BopN protein production and purification.** The recombinant 6×His-tagged BopN protein was produced in the *E. coli* BL21 $\lambda$(DE3) cells transformed with the pET28b-*Bp*Tohamal-BopN plasmid construct. The bacterial cells were grown in LB broth at 37°C supplemented with 60 $\mu$g/mL kanamycin. Expression of the protein was induced by the addition of 0.5 mM isopropyl 1-thio-$\beta$-d-galactopyranoside (IPTG, Alexis) when the optical density at 600 nm reached 0.6 and cultures were further grown at 37°C for additional 4 h. The cells were washed in TN buffer (50 mM Tris-HCl, pH 8.0, 150 mM NaCl) and disrupted by ultrasonic processor (Misonix S-4000, Misonix). The cell lysate was centrifuged at 40,000 g at 4°C for 30 min and the supernatant was loaded on a Ni Sepharose 6 High Performance column (Cytiva) equilibrated with TN buffer. The column was extensively washed with TN buffer supplemented with 50 mM imidazole and the BopN protein was eluted with TN buffer supplemented with 200 mM imidazole. The protein fractions were pooled and dialyzed against 50 mM Tris-HCl (pH 8), 150 mM NaCl at 4°C for 16 h. The dialyzed BopN protein was diluted with TN buffer to a concentration of 1 mg/mL and incubated with trypsin (Sigma) at molar ratio of 1:1,000. After 1 h of incubation at 25°C, the mixture was loaded on Ni Sepharose 6 High Performance column (Cytiva) equilibrated with TN buffer. The flow-through fractions containing the truncated BopN protein (BopN$_{69-365}$) were concentrated by ultrafiltration using a 10-kDa cutoff membrane (Amicon) and loaded on a Superdex 75 HR gel filtration column (GE Healthcare) equilibrated with 10 mM Tris-HCl (pH 7.4) and 150 mM NaCl. The eluted proteins were concentrated by ultrafiltration and stored at 4°C for further use. The purity of protein samples was monitored by SDS-PAGE. Protein concentrations were determined by Bradford assay (Bio-Rad) using bovine serum albumin (Sigma) as a standard.

**Crystallization of the BopN protein.** Crystallization screening was performed in a sitting-drop vapour-diffusion setup using a Gryphon (Art Robbins Instruments) and Oryx4 (Douglas Instruments Ltd) crystallization robots in MRC 2 Well (Hampton Research [HR]) and Combi Clover Junior crystallization plates (Rigaku Reagents), respectively. The diffraction quality BopN crystals with the dimensions of 350 $\times$ 50 $\times$ 30 $\mu$m grew within a week at 4°C in 0.16 M calcium acetate, 0.08 M sodium cacodylate, 14.4% (wt/vol) PEG 8000, 20% (vol/vol) glycerol employing a protein stock concentration of 8 mg/mL and protein-to-precipitant ratio 2:1.

**Data collection and processing.** Diffraction data were collected from an individual crystal at 100 K using synchrotron radiation on the MX14.1 beamline at BESSY II synchrotron (Helmholtz-Zentrum) equipped with the Pilatus3 S 2M detector (Dectris). 1800 diffraction images were collected with 0.1 oscillation angle. Collected data were processed using XDS (65) in XDSAPP interface (66).

**Structure solution and refinement.** The search model was generated by ARCIMBOLDO light (67, 68), based on the combination of locating small model fragments with density modification with the program SHELXE (69). The protein structure was solved by automated model-building method BUCCANEER (70). The structure was refined by REFMAC5 (71) and manually modeled using COOT (72). MolProbity server (73) was used for final model geometry validation. For the determination of the protein assembly, PDBePISA (74) was applied. The complete information about the data collection and refinement statistics is provided in Table 1. Coordinates and structure factors for the BopN protein were deposited in the Protein Data Bank under the accession code 7YYG.

**Determination of intracellular and secreted BopN$^{rep}$ and BteA$^{rep}$ levels by luminescence measurements.** To determine calcium-induced BopN secretion, *bopN*$^{rep}$ reporter bacteria were grown to exponential phase (OD$_{600}$ = 1.2) in calcium-rich SSM (2 mM Ca$^{2+}$) and transferred into SSM medium with different concentrations of Ca$^{2+}$. Following incubation for 90 min at 37°C, the cultures were centrifuged (10 min; 14,000 g) and amounts of BopN$^{rep}$ in supernatants (secreted BopN$^{rep}$) and extracts of cell pellets (intracellular BopN$^{rep}$) were determined (see below). To prepare the extracts, pellets were resuspended in 50 mM Tris-HCl, pH 8.0, and the cells were disrupted in 2 beating cycles of 3 min with 0.1 mm glass beads (Scientific Industries) using the Disruptor Genie (Scientific Industries). The extracts were clarified by centrifugation (10 min; 14,000 g).

For analysis of secreted and intracellular BteA amounts, reporter bacteria of *bteA*$^{rep}$ grown to exponential phase (OD$_{600}$ = 1.2) in calcium-rich SSM (2 mM Ca$^{2+}$) were inoculated to an OD$_{600}$ of 0.15 into

50 mL of SSM medium with or without 2 mM $Ca^{2+}$ and grown for various times at 37°C. One-mL aliquots of the cultures were taken after 3 h, 6 h, 9 h, and 24 h of growth. No difference in $OD_{600}$ was observed between cultures in SSM with or without 2 mM $Ca^{2+}$. After aliquot centrifugation (10 min; 14,000 g), amounts of BteA^rep in supernatants and cell extracts were assessed with Nano-Glo HiBit system.

Nano-Glo HiBiT Extracellular Detection System (Cat. No. N2420, Promega) was used, according to manufacturer's instructions. Briefly, supernatants and cell extracts were mixed with recombinant LgBit protein and furimazine substrate in Nano-Glo buffer, and lluminescence was measured using the FLUOstar Omega microplate reader (BMG LABTECH) or the TecanSpark microplate reader (Tecan). The purified recombinant protein BopN^rep and BteA^rep were used for the calibration curve and calculation of protein amounts.

**Production and purification of recombinant BopN^rep and BteA^rep.** The recombinant GST-tagged BopN^rep (aa 83 to 365 of BopN with C-terminal HiBit-3xFLAG-SPOT peptide) and BteA^rep (aa 1 to 130 of BteA with C-terminal HiBit peptide) were produced in *E. coli* XL-1 from pGEX-6P1 expression vector (GE Healthcare). Exponential *E. coli* cultures grown at 30°C were induced for protein production by adding IPTG to 0.1 mM at $OD_{600}$ = 0.3 and grown for an additional 16 h at 25°C. Bacterial cells were harvested by centrifugation, and the cell pellet was resuspended in ice-cold 50 mM Tris-HCl pH 7.4, 150 mM NaCl, and Complete Mini protease inhibitors (EDTA free, Roche). Bacterial cells were disrupted by ultrasound, and the lysate was clarified by centrifugation (20,000 g, 30 min). The recombinant proteins were purified from the supernatant fraction using columns prepacked with Glutathione-Sepharose 4B (Amersham). The resin with bound proteins was washed with 50 mM Tris-HCl pH 7.4, 150 mM NaCl and proteins were eluted with 10 mM reduced glutathione in 50 mM Tris-HCl pH 7.4, 150 mM NaCl. Protein preparations were dialyzed overnight into 50 mM Tris-HCl pH 7.4 and 150 mM NaCl. The integrity and purity of recombinant proteins were verified by SDS-PAGE electrophoresis followed by Coomassie blue staining, and protein concentration was determined by Bradford assay (Bio-Rad) using bovine serum albumin (Sigma) as a standard.

**Immunofluorescence staining and structured illumination microscopy.** Following centrifugation (5 min; 8,000 g) to remove SSM medium containing 2 mM $Ca^{2+}$, bacteria were re-inoculated into SSM medium with or without 2 mM $Ca^{2+}$. After incubation for 90 min at 37°C, cultures were spotted onto high-precision coverslips (#1.5 H), which were cleaned in 1N HCl and coated with 0.01% poly-L-lysine before use. Bacteria were allowed to adhere for 30 min and then fixed with 4% PFA in phosphate-buffered saline (PBS) for 75 min at RT. After washing with PBS, cells were permeabilized with 0.2% Triton X-100 in PBS at RT for 45 min and then washed in PBS with 0.05% Tween 20 (PBST). Blocking was performed with 4% BSA in PBST for 1 h, and primary antibodies were applied in 1% BSA in PBST at the following dilutions: SPOT-Label ATTO594 (Chromotek) at 1:600, and rabbit anti-*B. pertussis* serum (generously provided by Dr. Vecerek, Institute of Microbiology, Prague, Czech Republic) at 1:1,000. After incubation for 2 h at 37°C and washing with PBST (3 × 5 min), the secondary goat anti-rabbit antibody labeled with DyLight 405 (Jackson Immunoresearch) was applied in 1% BSA in PBST at the 1: 600 dilution. After incubation for 30 min at 37°C, Vectashield (H-1000-10, Vector Laboratories) was used to mount the coverslips containing the samples onto glass slides.

The DeltaVision OMX imaging platform was used to acquire images with 3D structured illumination microscopy (SIM). The system was equipped with the PLAN APO N 60x oil objective, N.A. 1.42; FWD 0.15; CG 0, four PCO and an Edge 5.5 sCMOS camera (readout speeds 95 MHz, 286 Mhz, 15 bit, pixel size: 6.5 $\mu$m). For excitation of fluorescent proteins and the labels on the antibodies, 405 nm diode, 488 nm OPSL, and 564 nm OPSL were used in combination with emission filters 435.5/31, 528/48, and 609/37 nm. SoftWoRx software (Applied Precision) was used for image reconstruction, deconvolution, and registration. Image processing, which consisted of cropping and brightness/contrast adjustment, was performed in FIJI ([75], NIH Bethesda), and final images were assembled in Adobe Illustrator (Adobe).

**Analysis of intracellular and secreted proteins by immunoblotting.** Bacteria of *bteA*^rep strains were grown in the SSM medium with and without 2 mM $Ca^{2+}$ overnight at 37°C. Alternatively, *bopN*^rep bacteria grown in calcium-rich medium (2 mM $Ca^{2+}$) were transferred into medium with various $Ca^{2+}$ concentrations (0, 0.1, 0.5, 1, and 2 mM) and incubated for 90 min. For analysis of intracellular protein content, bacterial cultures were centrifuged (30 min; 30,000 g) and pellets were lysed in 8M urea and 50 mM Tris-HCl, pH 8.0. Protein extracts were clarified by centrifugation (5 min; 14,000 g) and mixed with SDS-PAGE sample loading buffer. To analyze secreted protein content, culture supernatants were precipitated with 10% trichloroacetic acid overnight at 4°C, washed with acetone, dissolved in 8M urea, 50 mM Tris-HCl, pH 8.0, and mixed with SDS-PAGE sample loading buffer. Samples with ODs equivalent to 0.1 $OD_{600}$ unit (whole-cell lysates) or 1 $OD_{600}$ unit (bacterial supernatants) were separated by SDS-PAGE electrophoresis and transferred onto a nitrocellulose membrane. Membranes were probed overnight with mouse polyclonal antibodies raised against BopN (dilution 1:10,000), Bsp22 (dilution 1:10,000), BopD (dilution 1:10,000), or BopB (dilution 1:10,000), all kindly provided by Branislav Vecerek, Institute of Microbiology, Prague, Czech Republic. The detected proteins were revealed with 1:3,000-diluted horseradish peroxidase (HRP)-conjugated anti-mouse IgG secondary antibodies (GE Healthcare) using a Pierce ECL chemiluminescence substrate (Thermo Fisher Scientific) and an Image Quant LAS 4000 station (GE Healthcare).

**Determination of leakage of BteA^rep during infection.** To assess the amount of BteA^rep that leaked into the medium during cell infection, HeLa cells (1 × $10^6$ per well) in DMEM-10% FBS were seeded in a 12-well plate and allowed to adhere overnight. Bacteria of *bteA*^rep strains grown in calcium-rich SSM (2 mM $Ca^{2+}$) were washed in DMEM-10% FBS by centrifugation (5 min; 8,000 g) and added at MOI 5:1. The plate was centrifuged (5 min, 120 g) to allow efficient infection. Three hours after infection at 37°C

and 5% $CO_2$, aliquots of the medium were collected and clarified by centrifugation at 14,000 g for 10 min. The Nano-Glo HiBiT Extracellular Detection System (Cat.No. N2420, Promega) was used to determine the amount of BteA$^{rep}$ in the aliquots. Purified recombinant BteA$^{rep}$ protein was used for calibration curve and calculation of protein amounts.

**Determination of cellular injection of BopN$^{rep}$ and BteA$^{rep}$.** To assess the cellular injection of BopN$^{rep}$ and BteA$^{rep}$, LgBit-expressing HeLa cells ($5 \times 10^4$ per well) in DMEM-10% FBS were seeded into 96-well white/clear bottom plate (Corning) and allowed to adhere overnight. The Nano-Glo Live Cell Assay System (Cat.No. N2011, Promega) was employed to determine the amounts of injected BopN$^{rep}$ or BteA$^{rep}$, according to (48). Bacteria of *bteA*$^{rep}$ and *bopN*$^{rep}$ strains grown in calcium-rich SSM (2 mM $Ca^{2+}$) were washed in DMEM-10% FBS by centrifugation (5 min; 8,000 g) and added at MOI 5:1 along with cell-permeable luciferase substrate and Nano-Glo buffer. After centrifugation (5 min, 120 g), the plate was placed inside the chamber of TecanSpark microplate reader (Tecan) with 37°C and 5% $CO_2$ and luminescence measurements were performed for 2 h at 5 min intervals.

**Determination of NF-$\kappa$B activation.** To assess the BopN-mediated modulation of the NF-$\kappa$B pathway, A549 Dual reporter cells encoding secreted embryonic alkaline phosphatase (SEAP) under the control of the IFN-$\beta$ minimal promoter fused to five NF-$\kappa$B binding sites were used. Cells ($2 \times 10^4$ per well) in DMEM-10% FBS were seeded in 96-well plates and allowed to attach overnight. The derived mutant strains of *B. bronchiseptica* RB50 were added at the indicated MOI and centrifuged (5 min, 120 g). After incubation at 37°C and 5% $CO_2$ for 20 h, the amount of SEAP in cell culture supernatants was determined using QUANTI-Blue detection reagent (rep-qbs, Invivogen) according to the manufacturer's instructions. The amount of SEAP in the supernatant of cells stimulated with 1 ng/mL TNF-$\alpha$ was taken as 100%.

**Cytotoxicity assay.** Cytotoxicity of *B. bronchiseptica* toward A549 cells and Raw 264.7 macrophages was determined as changes in cell membrane integrity using the fluorescent DNA-binding dye CellTox Green (Cat. No. G8743, Promega). In brief, $2 \times 10^4$ A549 cells or $2.5 \times 10^4$ Raw 264.7 macrophages per well were seeded in a 96-well black/clear bottom plate (Corning) in DMEM-10% FBS and allowed to adhere overnight. *B. bronchiseptica* and derived mutant strains were added at the indicated MOI along with CellTox Green. The plate was then centrifuged (5 min, 120 g) and placed inside the chamber with 37°C and 5% $CO_2$ of the TecanSpark microplate reader (Tecan). Fluorescence measurements at 490ex/525em were performed at 15-min intervals for 20 h.

**Statistical analysis.** The significance of the differences between groups was determined by unpaired two-tailed $t$ test. Differences were considered statistically significant at $P < 0.01$.

## SUPPLEMENTAL MATERIAL

Supplemental material is available online only.

**SUPPLEMENTAL FILE 1**, PDF file, 2.8 MB.

## ACKNOWLEDGMENTS

This work was supported by the grant 21-05466S of the Czech Science Foundation (www.gacr.cz) and the Lumina Queruntur Fellowship LQ200202001 of the Czech Academy of Sciences to J.K., and the project National Institute of Virology and Bacteriology Program EXCELES, ID Project No. LX22NPO5103 - Funded by the European Union - Next Generation EU of the Ministry of Education, Youth, and Sports of the Czech Republic (https://www.msmt.cz) to P.S.

We also acknowledge the support from the Institutional Research Concept of Institute of Microbiology of the Czech Academy of Sciences RVO 61388971, project LM2023053 (Czech National Node to the European Infrastructure for Translational Medicine) from Ministry of Education, Youth and Sports of the Czech Republic, and the Light Microscopy Core Facility, especially Ivan Novotny, IMG, Prague, Czech Republic, supported by MEYS (LM2018129, CZ.02.1.01/0.0/0.0/18_046/0016045) and RVO–68378050-KAV-NPUI, for their support with the superresolution imaging presented herein.

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
