## [Reviewer comments · Microbiology Spectrum]

Microbiology Spectrum

BopN is a gatekeeper of the *Bordetella* type III secretion system

Kevin Navarrete, Ladislav Bumba, Tatyana Prudnikova, Ivana Malcova, Tania Allsop, Peter Sebo, and Jana Kamanova

Corresponding Author(s): Jana Kamanova, Mikrobiologicky ustav Akademie ved Ceske republiky

Review Timeline:

Submission Date:	October 11, 2022
Editorial Decision:	November 28, 2022
Revision Received:	January 24, 2023
Editorial Decision:	February 9, 2023
Revision Received:	March 14, 2023
Accepted:	March 17, 2023

Editor: Eric Cascales

Reviewer(s): The reviewers have opted to remain anonymous.

Transaction Report:

DOI: <https://doi.org/10.1128/spectrum.04112-22>

November 28, 2022

Dr. Jana Kamanova
Institute of Microbiology, Czech Academy of Sciences
Laboratory of Infection Biology
Videnska 1083
Prague 4 142 20
Czech Republic

Re: Spectrum04112-22 (BopN is a gatekeeper of the *Bordetella* type III secretion system)

Dear Dr. Kamanova:

Thank you for submitting your manuscript to Microbiology Spectrum. Your manuscript has been sent to two experts in the field. As you will see in their comments below, the two reviewers noticed that you report interesting findings. I have however received conflicting recommendations. While reviewer #2 is very positive about your work, reviewer #1 raised a significant number of issues notably regarding additional control experiments, or normalization of the data. I therefore encourage you to carefully address the reviewer's concerns and I invite you to submit a revised version. When submitting the revised version of your paper, please provide (1) point-by-point responses to the issues raised by the reviewers as file type "Response to Reviewers," not in your cover letter, and (2) a PDF file that indicates the changes from the original submission (by highlighting or underlining the changes) as file type "Marked Up Manuscript - For Review Only". Please use this link to submit your revised manuscript - we strongly recommend that you submit your paper within the next 60 days or reach out to me. Detailed instructions on submitting your revised paper are below.

Link Not Available

Sincerely,

Eric Cascales

Journals Department
Reviewer comments:

Reviewer #1 (Comments for the Author):

In this manuscript, the authors studied the BopN protein of *Bordetella*, which so far was considered a T3SS effector, and suggest that based on its structure and function, it is the unidentified gatekeeper protein of the *Bordetella* T3SS. While the results of this study are exciting and support the notion that BopN is involved in BteA secretion and its ability to translocate into the host cells' cytoplasm, they do not support the classical function of a gatekeeper. Therefore, I recommend reconsidering the presentation of BopN as the gatekeeper protein as well as addressing the following concerns:

Main comments:

1. A gatekeeper protein is a protein that switches the substrate secretion from the secretion of translocators to the secretion of effectors. In Fig. 2, the authors report the secretion of BopN, which is not a classical characteristic of a gatekeeper protein. Fig. 3 shows the secretion of the BteA effector, which is more relevant; therefore, the authors should change the order of the figures and present the results that demonstrate the effect of BopN on effector secretion first.
2. Figure 2C - The presentation of BopN amounts as a fraction of that expressed in a specific culture (BopNrep grown with 2 mM calcium) is not ideal. As it looks like the growing conditions affect the expression of BopNrep, it should be presented as a fraction of each sample (how much is secreted vs. intracellular). Also, please change the color of the black bars to a lighter color so that the error bars will be visible.
3. In Fig. 3B - the normalization of BteA secretion/expression is normalized according to the levels in the culture grown without calcium, while in Fig. 2B, it is normalized to a culture grown in 2 mM calcium. It is not clear why. Since this normalization method is not ideal, as discussed in my previous point, I think it should be presented as a ratio between each sample's secreted and intracellular levels.
4. Figure 2D-E - to have a complete picture of the localization BopN under high and low calcium concentrations, the authors should add the BscD localization for zero calcium and its merged image (so the two panels will become one).
5. Figure 3B-C - In Fig. 3B, the secretion/expression is measured under exponential conditions, while in Fig. 3C, it is examined for overnight cultures (stationary phase). To conclude the effect of BopN on effector/translocator secretion - the authors should present data from similar conditions or at least provide a scientific explanation for the difference.
6. Figure 4C - please add a positive control (WT strain or delta BopN strain) to show that NF- κ B activity is enhanced under these conditions.
7. The hierarchy of type III secretion should be explained in the introduction and not in the discussion (lines 333-343), as it is critical for understanding the work described in the manuscript.
8. Can you detect BopN-StcV interaction in vitro? Is it calcium-dependent?
9. Usually, gatekeeper proteins are not secreted. Here the authors describe a unique situation where BopN functions as a gatekeeper but is also secreted. Since its role as a gatekeeper requires the protein to be folded, the authors should explain how it later/simultaneously gets secreted. As a folded protein? How is this possible based on the reported structure?
10. Lines 182-185: the authors cannot conclude that the secretion of BopN is T3-dependent because it is not secreted in the delta BscN mutant, as it is hardly expressed in this strain, as mentioned by the authors in lines 186-188.
11. The authors should include the statistical analysis in their figures and clearly mark significant differences.
12. While the structural data presented in this manuscript was BopN of *B. pertussis*, the functional experiments were carried out with *B. bronchiseptica*. Can the authors add sequence alignment of the BopN of the two strains to determine that these proteins are homologous?
13. The design of BopN includes a flag-tag, which allows a direct measurement of BopN expression and secretion using western blot. Therefore, the authors should add this method to figures 2 and 3.
14. The authors should demonstrate that the BopN-HiBit-3xFlag-SPOT is a functional protein (preferably by showing that the labeled protein can complement the infection activity of the delta BopN mutant strain).
15. According to the Dali algorithm, "strong matches" have a Z-score higher than $n/100-4$ (where n is the number of amino acids). When taking ~280 aa of the BopN, only a z-score above 24 is meaningful. However, the authors report Z-scores of 12.5-20.3 but suggest structural homology. This point should be discussed when concluding the similarity between BopN and other T3SS gatekeepers.

Minor comments:

1. "secretion-restrictive calcium conditions", "physiological concentration", "high calcium condition". Please use one term to avoid confusion.
2. The abbreviation T3E is used in many figures but is not explained. Please add the acronym "type III effector" to the figure legend of Figure 2.
3. Figure 3C - the authors should add the size marker of the blots.
4. Line 175, add the calcium concentration tested.
5. Line 155 - the authors present the calcium response as a hypothesis, although it was already reported. I would recommend rephrasing the sentence to "previously reported."
6. *Bordetella* should be italic throughout the manuscript (for example, lines 76, 148, etc.).

Reviewer #2 (Comments for the Author):

In this study, Munoz Navarrete and co-workers report an extensive structure, biochemical and functional characterisation of the *Bordetella* T3SS protein BopN. Notably, they demonstrate that it possess a fold similar to that of the canonical T3SS gatekeeper protein SctW, that it regulates T3SS in a calcium-dependent manner, and that it regulates effector secretion. Notably, through an elegant reporter assay, they demonstrate that it is secreted (unlike the gatekeeper in many other T3SSs), but contrary to previous reports, it does not appear to impact the NF- κ B signalling cascade, nor impacts the secretion of the translocon components.

Collectively the work reported here is very compelling and performed thoroughly. While there are no major surprises, this study

contributes to the characterisation of this critical component of the T3SS, whose role and mechanism of action remains poorly understood. Accordingly, I only have very minor comments.

- In the abstract, line 30: HU after parepetrussi is probably a typo
- Abstract line 32, and also introduction line 77: why "so-called"? These are effectors, and have been shown to be secreted in the host.
- Line 96: While I commend the use to the standard T3SS nomenclature throughout the manuscript, it would probably help the reader to mention the name of the orthologue in other species here (SepD/L in EPEC, MxiC in Shigella...)
- Line 145: What is the sequence identity between homologous proteins? They are indicated in table 2, which should be referred to here - but is the sequence identity for the whole protein, or just the aligned residues?
- In the discussion, with might be worth mentioning if BopN possesses a bonafide T3SS signal sequence. It has been shown to have one in some orthologues (SepL I think?), but still is not secreted in this species. Is there something different in the signal sequence of BopN?

Staff Comments:

Preparing Revision Guidelines

Please return the manuscript within 60 days; if you cannot complete the modification within this time period, please contact me. If you do not wish to modify the manuscript and prefer to submit it to another journal, please notify me of your decision immediately so that the manuscript may be formally withdrawn from consideration by Microbiology Spectrum.

In this manuscript, the authors studied the BopN protein of *Bordetella*, which so far was considered a T3SS effector, and suggest that based on its structure and function, it is the unidentified gatekeeper protein of the *Bordetella* T3SS. While the results of this study are exciting and support the notion that BopN is involved in BteA secretion and its ability to translocate into the host cells' cytoplasm, they do not support the classical function of a gatekeeper. Therefore, I recommend reconsidering the presentation of BopN as the gatekeeper protein as well as addressing the following concerns:

Main comments:

1. A gatekeeper protein is a protein that switches the substrate secretion from the secretion of translocators to the secretion of effectors. In Fig. 2, the authors report the secretion of BopN, which is not a classical characteristic of a gatekeeper protein. Fig. 3 shows the secretion of the BteA effector, which is more relevant; therefore, the authors should change the order of the figures and present the results that demonstrate the effect of BopN on effector secretion first.
2. Figure 2C – The presentation of BopN amounts as a fraction of that expressed in a specific culture (BopN^{rep} grown with 2 mM calcium) is not ideal. As it looks like the growing conditions affect the expression of BopN^{rep}, it should be presented as a fraction of each sample (how much is secreted vs. intracellular). Also, please change the color of the black bars to a lighter color so that the error bars will be visible.
3. In Fig. 3B – the normalization of BteA secretion/expression is normalized according to the levels in the culture grown without calcium, while in Fig. 2B, it is normalized to a culture grown in 2 mM calcium. It is not clear why. Since this normalization method is not ideal, as discussed in my previous point, I think it should be presented as a ratio between each sample's secreted and intracellular levels.
4. Figure 2D-E – to have a complete picture of the localization BopN under high and low calcium concentrations, the authors should add the BscD localization for zero calcium and its merged image (so the two panels will become one).
5. Figure 3B-C – In Fig. 3B, the secretion/expression is measured under exponential conditions, while in Fig. 3C, it is examined for overnight cultures (stationary phase). To conclude the effect of BopN on effector/translocator secretion – the authors should present data from similar conditions or at least provide a scientific explanation for the difference.
6. Figure 4C – please add a positive control (WT strain or delta BopN strain) to show that NF-κB activity is enhanced under these conditions.
7. The hierarchy of type III secretion should be explained in the introduction and not in the discussion (lines 333-343), as it is critical for understanding the work described in the manuscript.
8. Can you detect BopN-StcV interaction in vitro? Is it calcium-dependent?
9. Usually, gatekeeper proteins are not secreted. Here the authors describe a unique situation where BopN functions as a gatekeeper but is also secreted. Since its role as a gatekeeper requires the protein to be folded, the authors should explain how it later/simultaneously gets secreted. As a folded protein? How is this possible based on the reported structure?
10. Lines 182-185: the authors cannot conclude that the secretion of BopN is T3-dependent because it is not secreted in the delta BscN mutant, as it is hardly expressed in this strain, as mentioned by the authors in lines 186-188.

11. The authors should include the statistical analysis in their figures and clearly mark significant differences.
12. While the structural data presented in this manuscript was BopN of *B. pertussis*, the functional experiments were carried out with *B. bronchiseptica*. Can the authors add sequence alignment of the BopN of the two strains to determine that these proteins are homologous?
13. The design of BopN includes a flag-tag, which allows a direct measurement of BopN expression and secretion using western blot. Therefore, the authors should add this method to figures 2 and 3.
14. The authors should demonstrate that the BopN-HiBit-3xFlag-SPOT is a functional protein (preferably by showing that the labeled protein can complement the infection activity of the delta BopN mutant strain).
15. According to the Dali algorithm, "strong matches" have a Z-score higher than $n/100-4$ (where n is the number of amino acids). When taking ~280 aa of the BopN, only a z-score above 24 is meaningful. However, the authors report Z-scores of 12.5-20.3 but suggest structural homology. This point should be discussed when concluding the similarity between BopN and other T3SS gatekeepers.

Minor comments:

1. "secretion-restrictive calcium conditions", "physiological concentration", "high calcium condition". Please use one term to avoid confusion.
2. The abbreviation T3E is used in many figures but is not explained. Please add the acronym "type III effector" to the figure legend of Figure 2.
3. Figure 3C – the authors should add the size marker of the blots.
4. Line 175, add the calcium concentration tested.
5. Line 155 – the authors present the calcium response as a hypothesis, although it was already reported. I would recommend rephrasing the sentence to "previously reported."
6. Bordetella should be italic throughout the manuscript (for example, lines 76, 148, etc.).

Response to reviewers of the manuscript Spectrum04112-22:

Reviewer #1:

Recommendation: While the results of this study are exciting and support the notion that BopN is involved in BteA secretion and its ability to translocate into the host cells' cytoplasm, they do not support the classical function of a gatekeeper. Therefore, I recommend reconsidering the presentation of BopN as the gatekeeper protein as well as addressing the following concerns.

A: We thank the reviewer for critical reading and the number of valuable comments, including the recommendation to reconsider the presentation of BopN as a gatekeeper protein. However, as discussed below, we politely disagree with such a strict definition of a gatekeeper, as the properties of gatekeeper proteins vary widely between different bacteria. In our opinion, the only defining functional feature of this class of proteins is the control over effector secretion/translocation. Therefore, we prefer to keep presenting BopN as a gatekeeper protein. To handle the suggestion of the reviewer and avoid confusion of the reader, we have now highlighted the similarities and differences of BopN with other members of this protein family in an improved texts of the Introduction and Discussion.

Main comments:

1. A gatekeeper protein is a protein that switches the substrate secretion from the secretion of translocators to the secretion of effectors. In Fig. 2, the authors report the secretion of BopN, which is not a classical characteristic of a gatekeeper protein. Fig. 3 shows the secretion of the BteA effector, which is more relevant; therefore, the authors should change the order of the figures and present the results that demonstrate the effect of BopN on effector secretion first.

A1: We followed the suggestion of the reviewer and reversed the order of the figures. However, as outlined above, we disagree with the statement that classical gatekeepers are not secreted. After activation, the fate of the gatekeeper varies from bacterial species to bacterial species. Among the examples of secreted/translocated gatekeepers are the *Shigella* MxiC (Botteaux A. *et al.* 2009, PMID: 19017268), *Yersinia* YopN (Cheng LW. *et al.* 2001, PMID: 11514512), *Pseudomonas* PopN (Yahr TL *et al.*

1997, PMID: 9371466), and *Chlamydia* CopN (Fields KA *et al.* 2002, PMID: 11123678). The SctW protein of *Aeromonas* AopN has also been shown to be secreted (Bergh PV *et al.* 2013, PMID: 24073886), although nothing is known about its gatekeeping function. It is currently unknown what determines whether the gatekeeper is secreted or not. One possible explanation could be the nature of its interactions within the type III secretion injectisome and/or the presence of a secretion signal, the prediction of which remains a challenge.

2. Figure 2C - The presentation of BopN amounts as a fraction of that expressed in a specific culture (BopNrep grown with 2 mM calcium) is not ideal. As it looks like the growing conditions affect the expression of BopNrep, it should be presented as a fraction of each sample (how much is secreted vs. intracellular). Also, please change the color of the black bars to a lighter color so that the error bars will be visible.

A2: The reviewer is correct that normalizing to protein levels in the specific culture may simplify the presentation and the recommendation was followed. However, the price for this type of normalization is the loss of the information on protein levels under specific culture conditions and/or during T3SS activation status. Indeed, the data show a downregulation of BopN levels in the T3SS-deficient $\Delta bscN$ strain that is independent of 2 mM calcium, as shown now in Fig. 3B. Furthermore, an increase in total BteA levels under secretion-promoting conditions with no effect on intracellular BteA levels was observed, as now shown in Fig. 2C. Therefore, on top of the performed normalization, shown in Fig. 3C, also absolute BopN levels from a representative experiment are now shown as the new Fig. 3B. Further, we replaced the color of the black bars with a lighter color to visualize the error bars.

3. In Fig. 3B - the normalization of BteA secretion/expression is normalized according to the levels in the culture grown without calcium, while in Fig. 2B, it is normalized to a culture grown in 2 mM calcium. It is not clear why. Since this normalization method is not ideal, as discussed in my previous point, I think it should be presented as a ratio between each sample's secreted and intracellular levels.

A3: As outlined above, the new Fig. 2 (previous Fig. 3) was modified according to the suggestion of the reviewer and data from a representative experiment were included.

4. *Figure 2D-E - to have a complete picture of the localization BopN under high and low calcium concentrations, the authors should add the BscD localization for zero calcium and its merged image (so the two panels will become one).*

A4: The suggestion was followed and the image of BscD/BopN localization for zero calcium was included into new Fig. 4 along with the corresponding fields from previous Fig. 2E. In the new Fig. 3 (previous Fig. 2), we show the merged images without the BscD protein to document the localization of the BopN protein in respect to the bacterial envelope, which is better seen in these images.

5. *Figure 3B-C - In Fig. 3B, the secretion/expression is measured under exponential conditions, while in Fig. 3C, it is examined for overnight cultures (stationary phase). To conclude the effect of BopN on effector/translocator secretion - the authors should present data from similar conditions or at least provide a scientific explanation for the difference.*

A5: The reviewer is correct that ideally the same conditions should be used for data shown in previous Fig. 3B and C, now Fig. 2. However, as documented in Suppl. Fig. 2, the ratios of intracellular and secreted BteA in WT and $\Delta bopN$ mutant cultures are consistently the same under exponential growth and stationary culture conditions. The immunoblot shown in Fig. 2E (previously Fig. 3C) was performed to examine whether ablation of BopN production affects the secretion of the Bsp22 and BopB/D structural subunits that are important for function of the injectisome. As shown, the secretion of the Bsp22 and BopB/D was not affected and we thus believe that use of higher amounts of these proteins accumulated in stationary culture for improved detection on the immunoblot is acceptable.

6. *Figure 4C - please add a positive control (WT strain or delta BopN strain) to show that NF-kB activity is enhanced under these conditions.*

A6: The suggestion of the reviewer could not be followed because under the conditions used, the BteA protein produced by wild-type and $\Delta bopN$ strains exerts such a potent cytotoxic action on A549 cells that the cells lyse before activation of the NF-kB pathway can be assessed by the used SEAP reporter system. In contrast, the $\Delta bteA$ strains are not cytotoxic and activation the NF-kB pathway in infected A549 cells could be assessed, showing that injection of BopN into cells does not inhibit NF-kB activation (now Fig. 5C). These data are now mentioned in the manuscript on lines 305-308.

7. *The hierarchy of type III secretion should be explained in the introduction and not in the discussion (lines 333-343), as it is critical for understanding the work described in the manuscript.*

A7: The suggestion was followed and the description of the hierarchy of the type III secretion is now included in the Introduction.

8. *Can you detect BopN-StcV interaction in vitro? Is it calcium-dependent?*

A8: The question of the reviewer is pertinent. In an ongoing project, we have unsuccessfully attempted to produce and purify the soluble cytosolic domain of BcrD, the SctV component of *Bordetella* injectisome. Therefore, we could not assess its interaction with BopN.

9. *Usually, gatekeeper proteins are not secreted. Here the authors describe a unique situation where BopN functions as a gatekeeper but is also secreted. Since its role as a gatekeeper requires the protein to be folded, the authors should explain how it later/simultaneously gets secreted. As a folded protein? How is this possible based on the reported structure?*

A9: The reviewer is correct that folded protein cannot be secreted by type III secretion injectisome. However, as mentioned above, it has already been shown that multiple other gatekeepers are secreted. It is assumed that during activation of effector secretion, either a mechanical or chemical signal disrupts gatekeeper interactions with its heteromeric chaperones and/or SctV proteins, leading in some cases to gatekeeper unfolding and secretion. Previously, the ATPase SctN was reported to be responsible for ATP-dependent release of chaperones and unfolding of the corresponding secreted proteins (Akeda Y *et al.* 2005, PMID: 16208377). However, its role in gatekeeper unfolding has not been investigated. To accommodate the question of the reviewer and clarify the issue for the reader, we have improved the text of the Discussion (lines 421-429).

10. *Lines 182-185: the authors cannot conclude that the secretion of BopN is T3-dependent because it is not secreted in the delta BscN mutant, as it is hardly expressed in this strain, as mentioned by the authors in lines 186-188.*

A10: We have modified the text accordingly.

11. *The authors should include the statistical analysis in their figures and clearly mark significant differences.*

A11: The statistical analysis was performed and significant differences are now included in Figure 2, Figure 3, Figure 5 and Figure 6.

12. *While the structural data presented in this manuscript was BopN of *B. pertussis*, the functional experiments were carried out with *B. bronchiseptica*. Can the authors add sequence alignment of the BopN of the two strains to determine that these proteins are homologous?*

A12: We are grateful to the reviewer for this pertinent comment as for the reader it may have been confusing that structure is given for *B. pertussis* BopN, while functional data are obtained with *B. bronchiseptica*. This could be done because there is a single P122T amino acid difference between the BopN proteins of *B. pertussis* Tohama I and *B. bronchiseptica* RB50, which maps into the loop between helix 3 and helix 4 of the structure. This information has now been indicated in the manuscript (lines 176-180). In fact, replacement of the T122 residue of *B. bronchiseptica* BopN by a proline residue in BopN of *B. pertussis* may potentially be among the reasons of the reduced functionality of the T3SS in *B. pertussis*, which will be subject of further examination. However, here we used the *B. pertussis* BopN structure only to reveal that *Bordetella* BopN has the characteristic three X-bundle domain architecture of T3SS gatekeepers, which prompted us to explore its gatekeeper function.

13. *The design of BopN includes a flag-tag, which allows a direct measurement of BopN expression and secretion using western blot. Therefore, the authors should add this method to figures 2 and 3.*

A13: The suggestion of the reviewer was followed only in part, as we have used mouse polyclonal antibodies raised against BopN for immunoblot detection of BopN, now shown in the new Fig. 3B. This method of detection is specific, as no signal is detected in the $\Delta bopN$ strain. Indeed, BopN detection by immunoblot was already included in the previous version of the Fig. 3C, now Fig. 2E.

14. *The authors should demonstrate that the BopN-HiBit-3xFlag-SPOT is a functional protein (preferably by showing that the labeled protein can complement the infection activity of the delta BopN mutant strain).*

A14: The data requested by the reviewer have been shown in another way in the initial manuscript by demonstrating that fusion of the HiBit-3xFlag-SPOT tag to BopN did not affect the BteA-dependent cytotoxicity of *B. bronchiseptica*, which would have been reduced upon loss of BopN function. This is now shown in Suppl. Fig. 3A, documenting that the *bopN^{rep}* reporter producing strain elicits the same T3SS-dependent cytotoxicity as the wild-type *B. bronchiseptica* RB50 strain. The issue is now also clarified in the manuscript.

15. According to the Dali algorithm, "strong matches" have a Z-score higher than $n/100-4$ (where n is the number of amino acids). When taking ~280 aa of the BopN, only a z-score above 24 is meaningful. However, the authors report Z-scores of 12.5-20.3 but suggest structural homology. This point should be discussed when concluding the similarity between BopN and other T3SS gatekeepers.

A15: The reviewer is correct that DALI defines a "strong match" between structures as an empirically determined Z-score higher than $(n/10)-4$, where n is the number of residues in the queried structure. Funnily enough, DALI also defines "significant similarities" to have a Z-score above 2. In any case, the Z-scores of BopN match well with the previously reported Z-scores within the SctW gatekeeper protein family. For example, DALI search of SepL of *E. coli* yields Z-scores between 11.9 and 7.6 for other gatekeeper proteins (Burkinshaw BJ *et al.* 2015, PMID: 26457522). The issue is now discussed on lines 161-168.

Minor comments:

1. "secretion-restrictive calcium conditions", "physiological concentration", "high calcium condition". Please use one term to avoid confusion.

A1: We apologize for the lack of consistency. The term has been unified to secretion restrictive conditions.

2. The abbreviation T3E is used in many figures but is not explained. Please add the acronym "type III effector" to the figure legend of Figure 2.

A2: The acronym has been added to the figure legend of Figure 2.

3. Figure 3C - the authors should add the size marker of the blots.

A3: We politely disagree with the reviewer, as only a section of the immunoblots comprising the detected proteins are shown in Fig. 3C, now Fig. 2E. Hence, adding size markers from entire immunoblots is impossible and makes little sense. To meet the presumed suggestion of the reviewer, we have indicated the detected Mw next to the protein name. We believe that this is fair, as the blot of culture supernatants comprises a rigorous negative control of the $\Delta bscN$ strain, which does not excrete any of the BopN, Bsp22, and BopB/D components. This demonstrates the high specificity of detection by the used antibodies.

4. *Line 175, add the calcium concentration tested.*

A4: The tested calcium concentrations have been added.

5. *Line 155 - the authors present the calcium response as a hypothesis, although it was already reported. I would recommend rephrasing the sentence to "previously reported."*

A5: We politely disagree with the reviewer. The chemical signals that activate effector secretion in different bacteria vary, ranging from high potassium concentration to low pH or low calcium concentration. The role of calcium concentration in *Bordetella* effector secretion has never been studied. Therefore, we hypothesized that low calcium concentration might serve as an artificial trigger for *Bordetella* effector secretion, and calcium-rich medium creates secretion-restrictive conditions. This hypothesis was tested and confirmed in the present paper.

6. *Bordetella should be italic throughout the manuscript (for example, lines 76, 148, etc.).*

A6: The line references used by the reviewer relate to the use of the term *bordetellae*, which is not a taxonomic description and thus should be written in regular font (e.g. Parkil J. *et al.* 2003, PMID: 12910271). We only use italics when referring to the genus *Bordetella*.

Reviewer #2:

Main comments:

1. *In the abstract, line 30: HU after parepetrussi is probably a typo*

A1: We thank reviewer, the typo has been corrected.

2. *Abstract line 32, and also introduction line 77: why "so-called"? These are effectors, and have been shown to be secreted in the host.*

A2: The words "so-called" have been removed.

3. *Line 96: While I commend the use to the standard T3SS nomenclature throughout the manuscript, it would probably help the reader to mention the name of the orthologue in other species here (SepD/L in EPEC, MxiC in Shigella...)*

A3: The suggestion is appreciated, and the introduction has been revised to provide more information on gatekeeper function, including names of orthologues in other species. Specifically, we have highlighted information on *Yersinia* YopN, while also other orthologues are also briefly mentioned.

4. *Line 145: What is the sequence identity between homologous proteins? They are indicated in table 2, which should be refereed to here - but is the sequence identity for the whole protein, or just the aligned residues?*

A4: The sequence identity of other BopN and YopN protein is now included in the introduction. The sequence identity in Table 2 represents the identity of the aligned residues. The issue is now clarified in the manuscript.

5. *In the discussion, with might be worth mentioning if BopN possesses a bonafide T3SS signal sequence. It has been shown to have one in some orthologues (SepL I think?), but still is not secreted in this species. Is there something different in the signal sequence of BopN?*

A5: We thank reviewer for pointing out the work on SepL (? Younis R. *et al* 2010, PMID: 20833800). To our knowledge, it is difficult to identify and predict signal sequences for T3SS substrates and we cannot say anything about the secretion signal of BopN. We believe that the different fate of SepL may be due to the interactions of its C-terminal parts with other T3SS components. Discussion of this issue is now included on lines 421-429 in the manuscript.

February 9, 2023

Dr. Jana Kamanova
Mikrobiologicky ustav Akademie ved Ceske republiky
Laboratory of Infection Biology
Videnska 1083
Prague 4 142 20
Czech Republic

Re: Spectrum04112-22R1 (BopN is a gatekeeper of the *Bordetella* type III secretion system)

Dear Dr. Kamanova:

Thank you for submitting your revised manuscript to Microbiology Spectrum. It has been sent back to the two original referees. Both acknowledge that you properly answered their initial comments and recommend publication. Reviewer #1 however suggests to add protein markers on the western-blots, and I agree with this remark. Per ASM style, it is required to have at least two weight markers on cropped blots. When submitting the revised version of your paper, please provide (1) point-by-point responses to the issues raised by the reviewers as file type "Response to Reviewers," not in your cover letter, and (2) a PDF file that indicates the changes from the original submission (by highlighting or underlining the changes) as file type "Marked Up Manuscript - For Review Only". Please use this link to submit your revised manuscript - we strongly recommend that you submit your paper within the next 60 days or reach out to me. Detailed instructions on submitting your revised paper are below.

Link Not Available

Sincerely,

Eric Cascales

Journals Department
Reviewer comments:

Reviewer #1 (Comments for the Author):

The authors nicely addressed all the critical points and provided sufficient scientific explanations for the issues that weren't clear. The only thing left is my request (as a minor comment) to add a size marker to the blots (presented in figures 2 and 3). In their rebuttal letter, the authors explain that since they show only a small section of the blots, it makes little sense to present a size marker. I beg to differ and argue that this is an elementary standard to show that the bands correspond to the expected protein size. I don't expect them to include the entire Mw markers (and I'm sorry I wasn't clear about it), but only one or two markers close to the bands. If the authors worry that it will clutter their presentation, I suggest they add the complete blots with the full Mw markers as supplementary material.

Reviewer #2 (Comments for the Author):

The authors have suitably addressed the comments from both reviewers, and I am happy to recommend publication of this revised manuscript in Spectrum.

Staff Comments:

Preparing Revision Guidelines

Please return the manuscript within 60 days; if you cannot complete the modification within this time period, please contact me. If you do not wish to modify the manuscript and prefer to submit it to another journal, please notify me of your decision immediately so that the manuscript may be formally withdrawn from consideration by Microbiology Spectrum.

Response to reviewers of the manuscript Spectrum04112-22R1 (BopN is a gatekeeper of the *Bordetella* type III secretion system)

Reviewer #1:

Q: The only thing left is my request (as a minor comment) to add a size marker to the blots (presented in figures 2 and 3). In their rebuttal letter, the authors explain that since they show only a small section of the blots, it makes little sense to present a size marker. I beg to differ and argue that this is an elementary standard to show that the bands correspond to the expected protein size. I don't expect them to include the entire Mw markers (and I'm sorry I wasn't clear about it), but only one or two markers close to the bands. If the authors worry that it will clutter their presentation, I suggest they add the complete blots with the full Mw markers as supplementary material.

A: We thank the reviewer for the explanation, and we have followed the recommendation. However, due to technical problems, the Western blots previously shown were not saved with the size marker in the correct format (overlay of the chemiluminescence signal with the brightfield view of the pre-stained marker). Therefore, we included a new set of Western blots in Figure 2 and Figure 3 and show their size markers in the supplemental material. The size markers of the blots from Figure 2 are shown in new Figure S3. The markers of the blots from Figure 3 are shown in revised Figure S4 (previous Figure S3).

Reviewer #2:

Q: The authors have suitably addressed the comments from both reviewers, and I am happy to recommend publication of this revised manuscript in Spectrum.

A: We thank the reviewer.

March 17, 2023

Dr. Jana Kamanova
Mikrobiologicky ustav Akademie ved Ceske republiky
Laboratory of Infection Biology
Videnska 1083
Prague 4 142 20
Czech Republic

Re: Spectrum04112-22R2 (BopN is a gatekeeper of the *Bordetella* type III secretion system)

Dear Dr. Jana Kamanova:

Thank you for submitting your revised manuscript and the efforts to provide the western-blot with the molecular weight markers. I am pleased to accept your manuscript for publication in Microbiology Spectrum. I am forwarding it to the ASM Journals Department for publication. You will be notified when your proofs are ready to be viewed.

Sincerely,

Eric Cascales
Editor, Microbiology Spectrum
